# Pre-Train Your Loss: Easy Bayesian Transfer Learning with Informative Priors

**Ravid Shwartz-Ziv**[*]
New York University
ravid.shwartz.ziv@nyu.edu

**Micah Goldblum**[*]
New York University
goldblum@nyu.edu

**Hossein Souri**
Johns Hopkins University

**Sanyam Kapoor**
New York University

**Chen Zhu**
University of Maryland

**Yann LeCun**
New York University
Meta AI Research

**Andrew Gordon Wilson**
New York University
andrewgw@cims.nyu.edu

## Abstract

Deep learning is increasingly moving towards a transfer learning paradigm whereby large foundation models are fine-tuned on downstream tasks, starting from an initialization learned on the source task. But an initialization contains relatively little information about the source task, and does not reflect the belief that our knowledge of the source task should affect the locations and shape of optima on the downstream task. Instead, we show that we can learn highly informative posteriors from the source task, through supervised or self-supervised approaches, which then serve as the basis for priors that modify the whole loss surface on the downstream task. This simple modular approach enables significant performance gains and more data-efficient learning on a variety of downstream classification and segmentation tasks, serving as a drop-in replacement for standard pre-training strategies. These highly informative priors also can be saved for future use, similar to pre-trained weights, and stand in contrast to the zero-mean isotropic uninformative priors that are typically used in Bayesian deep learning.

## 1 Introduction

The ability to transfer what is learned from one task to another — learning to ride a bicycle to then ride a unicycle — has historically set apart biological intelligence from machine learning approaches. However, *transfer learning* is quickly becoming mainstream practice in deep learning. Typically, large "foundation models" are pre-trained on massive volumes of source data, and then the learned parameter vector is used as an initialization for training in a downstream task. While this approach has had great empirical success, reliance on an initialization is a very limited way to perform transfer learning. If we are doing a good job of optimization, then our final solution should be independent of initialization, barring local minima with identical training loss. Moreover, our knowledge of the source task should affect the locations and shapes of optima on the downstream knowledge.

We propose to instead use a re-scaled Bayesian parameter posterior from the first task as a *pre-trained* prior for the downstream task. Since the negative log posterior on the downstream task is our loss function, this procedure has the effect of reshaping our training objective on the downstream task,

---

[*]Authors contributed equally.

36th Conference on Neural Information Processing Systems (NeurIPS 2022).

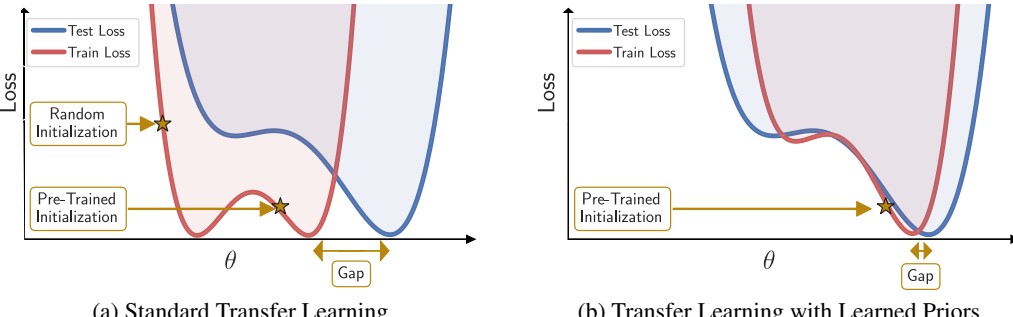

(a) Standard Transfer Learning      (b) Transfer Learning with Learned Priors

Figure 1: **Learned priors reshape the loss surface.** Red curves represent training loss, and blue curves represent the true population loss. The typical transfer learning approach avoids poor-generalizing minima by improving initializations (left). Learned priors further improve the alignment between training and test loss, improving accuracy (right). The gap between learned and optimal parameters is denoted by a double-sided gold arrow.

to reflect more nuanced information that we have learned from the source task, as in Figure 1. The posterior on the source task is re-scaled before use as a prior on the downstream task to reflect the belief that the source and downstream tasks are drawn from different distributions.

With our now highly informative prior for the downstream task, we can then proceed with optimization, or perform full Bayesian model averaging with the posterior on the downstream task. We find both procedures profoundly improve upon standard deep transfer learning, making use of a simple pipeline that involves easy-to-use pre-existing components. Indeed, the simplicity of the approach, combined with its promising empirical performance, is one of its greatest features — enabling its use as a drop-in replacement for standard approaches for deep transfer learning. The pre-training posterior can be saved as a prior for a wide array of downstream tasks, similar to pre-trained weights.

Despite the simplicity and effectiveness of this approach, Bayesian neural networks are almost always used with uninformative isotropic zero-mean priors [51, 25, 15], and have not harnessed recent advances in self-supervised learning. Furthermore, as we will show, the details of how we perform Bayesian transfer learning with modern neural networks are crucial to practical success. For example, it is important that the prior on the source task be represented with an informative covariance matrix, indicating which directions in parameter space we expect to provide good solutions on the source task, which significantly outperforms isotropic or diagonal counterparts. We also provide several key conceptual findings — for example, that the informative priors provide an inductive bias that enables more data efficient fine-tuning than a pre-trained initialization, and that pre-trained priors based on self-supervised learning are more transferable than supervised pre-trained priors. Finally, we observe through extensive experiments across both image classification and semantic segmentation on multiple neural architectures, and on both supervised and self-supervised (SimCLR [6]) pre-training loss functions, that Bayesian inference is particularly advantageous in the transfer learning setting.

We emphasize that the proposed approach outperforms standard transfer learning, without requiring any expert intervention or knowledge of Bayesian methods. One only needs to tune a single additional hyperparameter using standard cross-validation, which incurs minimal computational overhead. We release `PyTorch` pre-trained priors and code for learning and using priors for downstream inference: `https://github.com/hsouri/BayesianTransferLearning`.

## 2 Background

**Bayesian Neural Networks.** Consider a neural network $f$ with weights $w$ and a training dataset $\mathcal{D} = \{(x^{(i)}, y^{(i)})\}$ of independently drawn samples. In standard non-Bayesian neural network training, we minimize the negative log posterior, $-\log p(w|\mathcal{D}, f) \propto -\log p(\mathcal{D}|w, f) - \log p(w|f)$, where the likelihood $p(\mathcal{D}|w, f)$ denotes the probability that a model with weights $w$ would generate the observed labels $\{y^{(i)}\}$ (e.g., cross-entropy). The log prior $\log p(w|f)$ often takes the form of weight decay, corresponding to a zero-mean isotropic Gaussian prior. In Bayesian modeling,

we instead make predictions with a *Bayesian Model Average* (BMA) of all models weighted by their posterior probabilities: $p(y|x, \mathcal{D}, f) = \int p(y|x, w, f)p(w|\mathcal{D}, f)dw$. There are many ways to approximate this integral, such as MCMC, or variational approaches, which take a finite number of approximate samples $w_j$ from the parameter posterior to form the simple Monte Carlo estimate: $p(y|x, \mathcal{D}, f) \approx \frac{1}{J}\sum_j p(y|x, w_j, f)$. Almost always, one uses zero-mean isotropic priors [51]. There are also specialized priors, including heavy-tailed priors [15, 25], noise-contrastive priors for high uncertainty under distribution shift [17], and input-dependent priors for domain generalization [24].

**Transfer Learning.** In *transfer learning*, we wish to recycle the representation learned on one task to improve performance on another. Transfer learning is now widely applied in deep learning, and forms the basis for *foundation models* [1, 12, 8, 43, 4, 20, 10], which are exceptionally large neural networks pre-trained on massive volumes of data, and then fine-tuned on a downstream task. Recent work has found self-supervised pre-training can transfer better than supervised pre-training [19], in line with our finding in Section 4 that self-supervised pre-trained priors transfer better.

In *continual learning*, we wish to learn tasks sequentially without forgetting what has been learned previously. *Elastic Weight Consolidation* (EWC) prevents forgetting by imposing a penalty, the diagonal of a Fisher information matrix computed on previous tasks, for adapting to a new task [29]. We can contextualize the EWC penalty within Bayesian transfer learning as a negative log Gaussian prior with diagonal covariance used for maximum a posteriori (MAP) estimation on sequential tasks, such as digit classification or RL [29]. In our experiments, we find that MAP estimation, diagonal covariance, and supervised pre-training are all suboptimal in the transfer learning setting.

**Leveraging Auxiliary Data and Knowledge Transfer in Bayesian Modeling.** A number of works outside of deep learning have considered knowledge transfer in Bayesian modeling, especially in settings such as domain adaptation or homogeneous transfer learning in which source and target tasks contain similar feature and label spaces but differ in their marginal distributions. For example, Xuan et al. [52] considers Bayesian transfer learning methods for probabilistic graphical models, and Karbalayghareh et al. [28] develop a theoretical framework for understanding optimal Bayes classifiers given prior knowledge from other domains. Other works on Bayesian transfer learning learn a Dirichlet prior over naive Bayes classifiers [44]. Bayesian optimization methods can also find transferable hyperparameter settings [40] or select data which will yield transferable models on shifted domains [45]. Schnaus [47] uses the Laplace approximation to learn posteriors with Kronecker-factored covariance and then adjusts the posteriors by optimizing PAC-Bayes generalization bounds to create priors for downstream applications.

Bayesian tools have additionally been used for leveraging auxiliary data or multiple domains in deep learning. Chandra and Kapoor [2] learn from multiple domains simultaneously using a round-robin task sampling procedure and a single-layer neural network. Bayesian methods for continual learning update the posterior to accommodate new tasks without forgetting how to perform previous ones [35, 49, 13, 27, 46], or develop kernels based on neural networks trained on previous tasks for Gaussian process inference [33, 37]. Semi-supervised algorithms can incorporate unlabeled data into the training pipelines of BNNs using biologically plausible Bayesian Confidence Propagation Neural Networks (BCPNN) which model the cortex [41], by perturbing weights and using consistency regularization [11], or via semi-supervised deep kernel learning [26]. Gao et al. [16] also show how to harness unlabeled data to learn reference priors, uninformative priors which depend on the amount of training data and allow labels to most efficiently inform inference. Deep kernel learning has also been used for Bayesian meta-learning in few-shot classification and regression [39].

In contrast to these works, we do not perform multi-task learning nor is our goal to harness auxiliary unlabeled downstream data; we instead approach transfer learning in which we leverage pre-training data for expressive priors that maximize performance on a single downstream task.

## 3   Bayesian Transfer Learning with Pre-Trained Priors

In order to transfer knowledge acquired through pre-training to downstream tasks, we adopt a pipeline with three simple components, composed of easy-to-use existing tools:

1. First, we fit a probability distribution with closed-form density to the posterior over feature extractor parameters using a pre-trained checkpoint and SWAG [34] (Section 3.1).

2. Second, we re-scale this distribution, viewed as a prior for new tasks, to reflect the mismatch between the pre-training and downstream tasks and to add coverage to parameter settings which might be consistent with the downstream, but not pre-training, task. To this end, we tune a single scalar coefficient on held out validation data (Section 3.2).

   Finally, we use this re-scaled prior, along with a zero-mean isotropic Gaussian prior over added parameters (e.g. classification head), to form a posterior on the downstream task. We then either optimize the posterior, or use it to perform full Bayesian inference with SGLD and SGHMC samplers [42, 5] (Section 3.3)

3. Finally, we plug the re-scaled prior into a Bayesian inference algorithm, along with a zero-mean isotropic Gaussian prior over added parameters (e.g. classification head) to form a posterior on the downstream task. We then either optimize the posterior, or use it to perform full Bayesian inference with SGLD and SGHMC samplers [50, 5] (Section 3.3).

We illustrate this pipeline in Figure 5 in the Appendix. The simplicity and modularity of this framework are key strengths: by carefully combining easy-to-use existing components, we will see in Section 4 that we can straightforwardly improve the default approaches to deep transfer learning. The potential for impact is significant: we can use this pipeline as a drop-in replacement for standard procedures used in deploying foundation models. At the same time, there is significant novelty: while Bayesian neural networks are typically used with simple zero-mean isotropic priors, we leverage the significant developments in self-supervised learning to produce highly informative priors. Moreover, in the following subsections (3.1, 3.2, and 3.3) we will gain a nuanced understanding of each of these three components and the key considerations for practical success. We discuss computational considerations in Section 3.4.

For experiments in this section, we use a prior learned over the parameters of an ImageNet pre-trained SimCLR ResNet-50 feature extractor [9, 6, 18], and we choose CIFAR-10 and CIFAR-100 for downstream tasks [30]. We perform Bayesian inference with stochastic gradient Hamiltonian Monte Carlo (SGHMC) [5]. The experiments in this section are primarily intended to gain conceptual insights into each step of our approach. In Section 4, we present our main empirical evaluations.

### 3.1 Learning Transferable Priors

We begin by building a probability distribution over the parameters of a feature extractor which represents knowledge we acquire from pre-training. To this end, we fit the distribution to the Bayesian posterior, or regularized loss function, on the pre-training task. This pre-training posterior will become the prior for downstream tasks and can be saved, or publicly released, for future use, similar to pre-trained weights. This stage requires two major design choices: a pre-training task and an algorithm for constructing the probability distribution.

Our downstream inference procedures require that our prior is represented as a closed-form density function. Thus, we must use a method that can provide a closed-form posterior approximation for the source task, which can be then re-scaled and used as a prior in the downstream task. We opt for SWA-Gaussian (SWAG) [34] due to its simplicity, scalability, popularity, and non-diagonal covariance. Other procedures that provide closed-form posterior approximations, such as the Laplace approximation or variational methods, could also be applied, though as we will see, a non-diagonal covariance is particularly important to the success of this approach.

SWAG starts from a pre-trained model, and runs a small number $M$ of fine-tuning epochs with a modified learning rate schedule [34]. The SWAG approximate posterior distribution is given by $\mathcal{N}(\overline{w}, \frac{1}{2}\Sigma_{\text{diag}} + \frac{1}{2}\Sigma_{\text{low-rank}})$, where $\overline{w} = \frac{1}{M}\sum_{t=1}^{M} w_t$ is the SWA [23] solution, $\Sigma_{\text{diag}} = \frac{1}{L-1}\sum_{t=M-L+1}^{M} \text{diag}(w_t - \overline{w})$, and $\Sigma_{\text{low-rank}} = \frac{1}{L-1}\sum_{t=M-L+1}^{M} (w_t - \overline{w})(w_t - \overline{w})^{\top}$, where $L$ is a hyperparameter controlling the rank of the low-rank component of the covariance matrix. After obtaining a closed-form distribution using SWAG, we remove the head from on top of the feature extractor, for example, a linear classification module, and we consider only the distribution's restriction to the parameters of the feature extractor. New layers that are added for downstream tasks receive a non-learned prior over their parameters.

To highlight the versatility of Bayesian transfer learning, we will focus both on supervised image classification and also on self-supervised learning pre-training tasks and leverage existing `torchvision` and SimCLR [6] checkpoints. For both `torchvision` and SimCLR models, we learn the prior

using the associated loss function — cross-entropy and InfoNCE, respectively, regularized by weight decay. In both settings, the regularized loss function can be represented as the sum of a negative log-likelihood and the negative log Gaussian density (weight decay).

While standard SGD-based transfer learning uses a learned parameter vector, our Bayesian transfer learning approach uses a covariance matrix over feature extractor parameters which contains information about the pre-training loss surface geometry.

**Can we really "learn" a prior?** A data-dependent prior may sound odd, because a prior reflects our beliefs "before we see the data". However, it is entirely principled to learn a prior, as long as it is not learned using exactly the same labeled data we use in our downstream likelihood for the downstream task. Indeed, any informative prior is based on data that has shaped our beliefs.

**Alignment of Pre-Training and Downstream Loss Geometry.** If the covariance matrix of our learned prior is to be more effective than an isotropic one, then it must not favour poorly generalizing directions for the downstream task. Thus, we compare the alignment of the learned covariance matrix with a downstream task's test loss.

We begin by computing the top 5 leading singular vectors of a SWAG learned covariance matrix over parameters of the SimCLR ImageNet-trained ResNet-50 feature extractor. We then train a linear classifier head on CIFAR-10 training data on top of the fixed pre-trained feature extractor. Starting at the learned parameter vector, we perturb the feature extractor parameters in the direction of the singular vectors (fixing fully-connected classifier parameters), and measure the increase in test loss. We compare to the test loss when instead perturbing by each of 10 random vectors. All perturbation distances are filter normalized (as in [31, 21]), to account for invariance with respect to filter-wise parameter rescaling. In Figure 2a, we see the CIFAR-10 test loss is far flatter in the directions of leading eigenvectors of the pre-trained covariance than in a random direction, indicating the learned prior indeed promotes directions consistent with the downstream task.

**Learned Covariance Outperforms Only a Learned Mean.** After verifying that our pre-trained priors do in fact identify flat directions of the pre-training loss which are aligned with the downstream loss, we directly compare the performance benefits of our learned covariance over an isotropic covariance with a learned mean. To this end, we swap out our learned prior's covariance with an isotropic version $\alpha I$, where $\alpha$ is tuned on a held out validation set. Figure 2c shows that a learned covariance consistently outperforms its isotropic counterpart, indicating that the shape of the pre-training loss surface's basin is informative for downstream tasks. The $x$-axis here denotes the number of training samples used for fine-tuning on the downstream task. We also see Bayesian model averaging provides further performance gains, which we discuss further in Section 3.3. We now dissect just how important the low-rank component is.

**How is the Prior Best Structured?** In the context of continual learning, elastic weight consolidation (EWC) [29] uses purely diagonal covariance priors to help prevent forgetting in continual learning. In this paper, we are interested in transfer learning, and wish to understand the benefits of a low-rank component for capturing particularly important directions in the pre-training loss surface for transferring to downstream tasks. Omitting the low-rank component slightly reduces the memory footprint, but loses potentially important information about the shape of the pre-training posterior mode. In practice, the rank of the matrix is determined by the number of samples collected when running SWAG. In Figure 2b, we present the performance of a prior learned via SWAG on a SimCLR ResNet-50 feature extractor for transfer learning to CIFAR-10 and CIFAR-100. As the rank of the low-rank component increases from zero (diagonal covariance), we see the performance improves dramatically until it saturates, indicating that only a small number of dominant directions are important for the transferability of the prior. Note that performance saturation occurs later for CIFAR-100 than CIFAR-10 — likely due to the higher complexity of CIFAR-100 compared to CIFAR-10, which has $10\times$ fewer classes.

### 3.2 Rescaling the Prior

In *incremental learning*, it is common to acquire some data, form a posterior, and then use this posterior as our new prior in acquiring future data. This procedure is equivalent to forming the posterior from all of the data at once, assuming all data are drawn from the same distribution with the same likelihood. However, in transfer learning, we assume the data from the source and downstream task are drawn from *different* but related distributions. Thus we do not want to directly re-use a

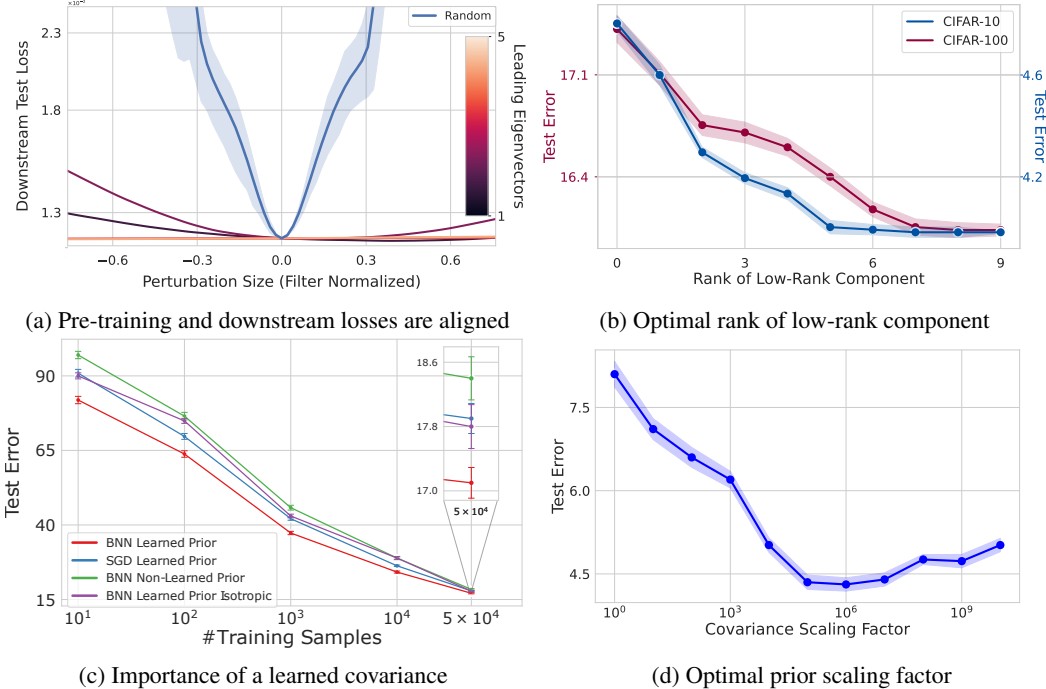

(a) Pre-training and downstream losses are aligned

(b) Optimal rank of low-rank component

(c) Importance of a learned covariance

(d) Optimal prior scaling factor

Figure 2: **Learning Transferable Priors.** Error bars corresponds to standard error over 5 trials. All experiments performed with ResNet-50 backbone pre-trained via SimCLR on ImageNet. **(a)** Test loss (CIFAR-10) is less sensitive to perturbations in the directions of leading singular vectors of a learned prior's covariance compared to random directions. Estimated over 10 random directions. **(b)** A non-diagonal covariance is important, though performance on downstream tasks plateaus at a relatively low rank. CIFAR-100 benefits from a higher rank covariance than CIFAR-10, due to its relative complexity. **(c)** A non-diagonal covariance outperforms an isotropic covariance with learned mean on CIFAR-100. Bayesian inference provides further performance gains. **(d)** We must re-scale the posterior from the source task, since data from each task is drawn from a different distribution. If $\lambda$ is too small, then we assume too much similarity between tasks. If $\lambda$ is too large, we lose useful information from the source task. Downstream inference is performed on CIFAR-10.

posterior from the source as a prior for the downstream task, without modification. For example, as we acquire more data from the source task, our posterior will become increasingly concentrated. This concentration is an issue, as certainty with respect to the ideal parameters for the pre-training task does not imply certainty on the ideal parameters for the downstream task.

To address this consideration, we rescale the learned Gaussian prior by multiplying its covariance matrix by a scalar value. We select the highest performing scalar value across a grid on a holdout set from the downstream task. If we do not scale the covariance enough, our prior will become concentrated around parameters which are inconsistent with the downstream task, and if we scale the covariance too much, our prior will be diffuse and assign too much mass to regions of parameter space which are again inconsistent with the downstream task.

We now put this intuition to the test. We fit the SimCLR pre-training loss using a Gaussian prior with mean $\mu$ and covariance matrix $\Sigma$. Given our uncertainty regarding the strength of the relationship between the pre-training and downstream tasks, it is unclear if our learned prior would lead to over-confidence on the downstream task. To rectify this problem, we instead assign prior covariance matrix $\lambda\Sigma$ with $\lambda \geq 1$. Figure 2d shows the accuracy of our method on CIFAR-10 as a function of $\lambda$. As expected, we see that we need to make the prior more diffuse if we are to optimize performance. In fact, prior rescaling can be the difference between strong and poor performance. We also see there is an optimal scaling factor where the prior is neither overly diffuse nor concentrated around a solution which is inconsistent with the downstream task. Additionally, small scaling factors constrain ourselves to poor parameters, hurting performance much more than large scaling factors, which simply inducing a near-uniform prior, negating the benefits of transfer learning.

### 3.3 Bayesian Inference

After learning a prior and re-scaling it, we finally must draw samples from the downstream task's posterior over the parameters of the entire model, including additional modules, such as classification heads, which were added after pre-training specifically for a particular downstream task. We use a zero-mean isotropic Gaussian prior over these additional parameters, with a scaling that is again tuned on held-out training data from the downstream task. Since we obtain a closed-form prior, the re-scaled distributions we learn are compatible with a wide variety of Bayesian inference algorithms. In our experiments, we choose stochastic gradient Hamiltonian Monte Carlo (SGHMC) [5] and Stochastic Gradient Langevin Dynamics (SGLD) [50] since these samplers simultaneously provide strong performance and tractable computation costs. Once we have obtained samples from the downstream posterior, we use these samples to form a Bayesian model average for test-time predictions.

In Figure 2c, we also observe the advantage of Bayesian inference (SGHMC) over MAP estimation (SGD) on the negative log-posterior yielded by our learned prior. Bayesian inference outperforms MAP estimation across all dataset sizes we consider.

In general, Bayesian inference can benefit most from an expressive prior. Indeed, priors reshape the whole posterior landscape — and the distinctive feature of a Bayesian approach is that it marginalizes over the whole posterior, rather than simply using an optimum as with standard training.

### 3.4 Practical Considerations

While our method requires a learned prior and a re-scaling coefficient, it is very easy to use and has minimal computational costs over standard fine-tuning routines. In particular, no expert intervention or knowledge of Bayesian methods is required, and only a single hyperparameter needs to be tuned. In the Appendix Section A, we evaluate the costs of each of the three stages of our pipeline in detail: (i) inferring the posterior on the source task; (ii) re-scaling the posterior to become an informative prior for the downstream task; (iii) using the informative prior in the downstream task.

In short, (i) has a minor cost (e.g., $\frac{1}{4}$ of an epoch on ImageNet) that is a one-time cost when a pre-trained posterior is publicly released, as we have done; (ii) involves a single hyperparameter that can be tuned in the same way as other hyperparameters (simple validation grid search), does not require specialized expertise, and adds $\frac{1}{7}$ to the runtime of fine-tuning; (iii) has no additional cost if we do MAP optimization and has a cost comparable to deep ensembles if we run our implementation of Bayesian inference, both in terms of fine-tuning and test-time, and we show that Bayesian inference significantly outperforms deep ensembles.

## 4 Experiments

We now conduct a thorough empirical evaluation of our transfer learning pipeline on image classification and semantic segmentation. We generally consider five approaches, which we find have the following performance ordering: **(1)** Bayesian inference with learned priors, **(2)** SGD with learned priors, **(3)** SGD with standard pre-training, **(4)** Bayesian inference with non-learned zero-mean priors, **(5)** SGD with non-learned zero-mean priors. We note that both (1) and (2) are part of our framework, and that SGD with learned priors (2) significantly outperforms standard transfer learning (3).

### 4.1 Experimental Setting

We adopt the ResNet-50 and ResNet-101 architectures [18], and we scale the input image to $224 \times 224$ pixels to accommodate these feature extractors designed for ImageNet data. We use a SimCLR (SSL) ResNet-50 checkpoint [6] pre-trained on the ImageNet 1k dataset [9] and fit our prior to the SimCLR loss function. For the supervised setting, we use pre-trained `torchvision` ResNet-50 and ResNet-101 backbones. We perform image classification experiments on four downstream tasks: CIFAR-10, CIFAR-100 [30], Oxford Flowers-102 [36], and Oxford-IIIT Pets [38]. On semantic segmentation, we use a DeepLabv3+ system [4] with ResNet-50 and ResNet-101 backbone architectures, and we evaluate performance on the Pascal VOC 2012 [14] and Cityscapes [7] datasets. All error bars represent one standard error over 5 runs. We evaluate over a variety of downstream training set sizes, as transfer learning often involves limited downstream data.We provide a detailed description of hyperparameters in Appendix C.1.

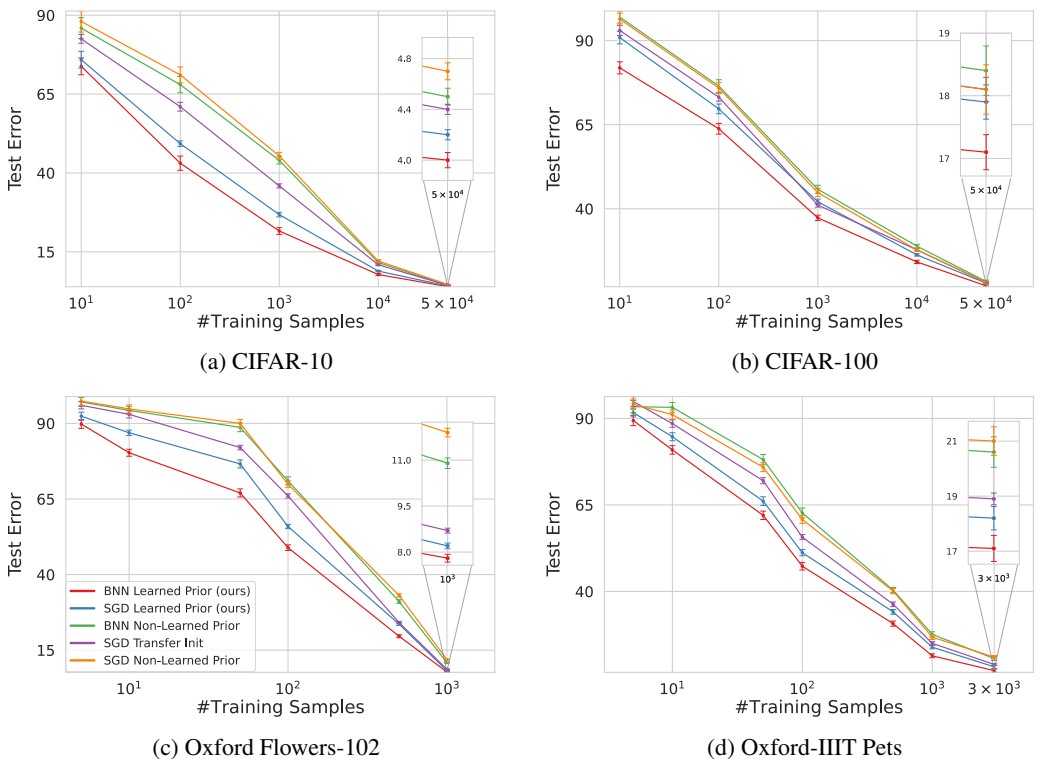

Figure 3: **Performance comparison.** Learned priors outperform non-learned priors, and Bayesian inference generally outperforms SGD training, especially with learned priors. Moreover, learned priors tend to be more data-efficient. We use a ResNet-50 and SimCLR for pre-trained priors. The horizontal axis denotes the number of downstream training samples on a logarithmic scale. Error bars correspond to a standard error over 5 trials.

## 4.2 Performance Comparison

In Figure 3, we compare the five approaches described above across various dataset sizes. We observe the following: (i) learned priors consistently outperform SGD transfer learning which in turn outperforms non-learned priors; (ii) conditioned on using the same prior, Bayesian inference often outperforms SGD training; (iii) Bayesian inference adds more value when used with informative priors; (iv) learned priors are relatively most valuable on intermediate data sizes for the downstream task. Point (iv) is particularly interesting: unlike standard SGD transfer learning, which involves an initialization from the pretraining task, the learned priors provide an explicit inductive bias, enabling the resulting model to learn more efficiently from downstream data. Once there is a sufficient amount of downstream data, the source pre-training becomes less important, and the methods become more similar in performance, although there is still a significant gap.

**Computation and comparison to deep ensembles.** Our Bayesian model average (BMA) contains 10 samples from the posterior, and thus incurs a higher test-time cost than a single SGD-trained model. We therefore additionally compare to an equally sized ensemble of SGD-trained transfer learning models in Appendix B.3. We see that our BMA strongly outperforms the deep ensemble. In transfer learning, deep ensemble members are initialized with the same pre-trained checkpoint, and fine-tuning tends to stay in the same basin, such that the ensemble components are relatively homogenous in their predictions. Recall that SGD with our learned priors incurs essentially the same computational costs as standard transfer learning, but has much better performance.

**Comparison between self-supervised and supervised pre-training.** In Appendix B.2 we provide additional evaluations with priors generated using `torchvision` ResNet-50 and ResNet-101 checkpoints. These priors differ from the SimCLR prior in that they are learned with labeled data rather than a self-supervision task. We see here also that learned supervised priors paired with Bayesian

inference consistently outperforms all baselines. We also see, notably, that the SSL priors are more transferable and outperform their supervised counterparts.

**Evaluating uncertainty.** We also measure predictive uncertainty via negative log test-likelihood (NLL) in Figure 4b, with additional results in Appendix B.4. We find that Bayesian transfer learning outperforms all other methods. However, we note that even though Bayesian inference with a non-learned prior has inferior accuracy to SGD with pre-training, it has superior test-likelihood — indicating that the confidence of SGD-based transfer learners is significantly miscalibrated, as likelihood accounts for both accuracy and uncertainty. We also evaluate the calibration of uncertainty using reliability diagrams (Figure 9 in the Appendix). These plots demonstrate that Bayesian inference with learned priors is the best calibrated among methods we consider. .

## 4.3 Out-of-Distribution Generalization

CIFAR-10.1 [42] is a test set of 2000 natural images modeled after CIFAR-10 with the same classes and image dimensions, following a similar dataset creation process to that described in the original CIFAR-10 paper. Models trained on CIFAR-10 consistently perform much worse on CIFAR-10.1 despite the images being similarly easy for humans to classify. Figure 4a indicates that our method achieves superior performance to SGD-based transfer learning and Bayesian inference with non-learned priors on CIFAR-10.1 across training set sizes, where training sets are sampled from CIFAR-10 training data.

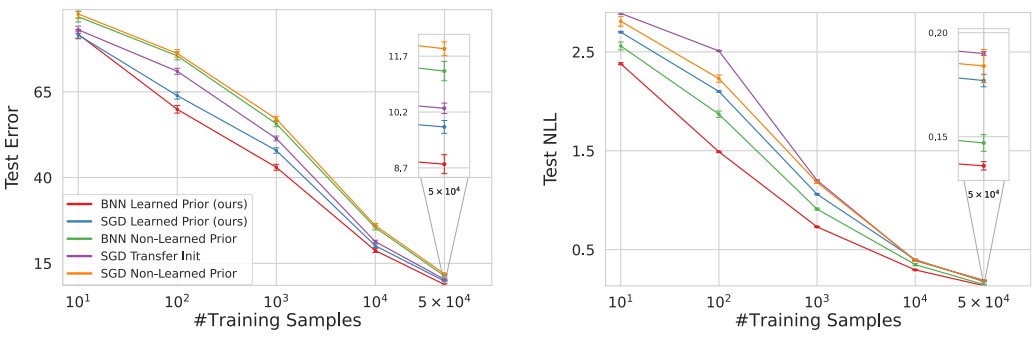

(a) Distributional shift on CIFAR-10.1.    (b) Test Negative Log-Likelihood on CIFAR-10.

Figure 4: **Predictive likelihood and distribution shift.** The horizontal axis denotes the number of downstream CIFAR-10 training samples on a logarithmic scale. Error bars correspond to standard error over 5 trials. All experiments performed with ResNet-50 backbone pre-trained via SimCLR on ImageNet. **(a)** BNNs with our learned priors similarly achieve superior accuracy on OOD testing. **(b)** Bayesian inference with a learned prior achieves superior test NLL, but BNNs with a non-learned prior outperforms all other methods despite often performing worst in terms of accuracy.

## 4.4 Semantic Segmentation with ImageNet Priors

Popular segmentation methods use ImageNet pre-trained weights as an initialization for backbone parameters [4]. We observe that semantic segmentation models, which contain numerous parameters aside from those of the backbone, still benefit immensely from a learned prior over backbone parameters. Placing a strong prior, even if over only the backbone component of the segmentation system, enables us to more fully harness pre-training and boost performance across benchmark datasets PASCAL VOC 2012 and Cityscapes.

When constructing a prior, we apply SWAG to `torchvision` ("Supervised") and SimCLR ("SSL") ResNet-50 models and a `torchvision` ResNet-101 model (the authors of SimCLR did not release a ResNet-101 model). We assign a zero-mean isotropic Gaussian prior to the decoder and atrous convolution parameters of DeepLabv3+ which are not pre-trained.

In Table 1, we see that Bayesian inference with learned priors achieves superior Mean-IoU to both baselines (SGD and SGLD) without learned priors. Priors learned on the SimCLR objective again outperform those learned on labeled data. We also present experiments with MAP estimates (SGD)

for the loss functions induced by both learned priors in Appendix C.2. We again find that our learned priors boost the performance of MAP estimates as well, and thus are preferable to standard transfer learning, even for practitioners who do not intend to perform Bayesian inference.

| Dataset | Backbone | SGD Transfer | Non-Learned Prior | Learned Prior Supervised | Learned Prior SSL |
|---|---|---|---|---|---|
| PASCAL VOC 2012 | ResNet-50 | 73.17 | 73.16 | 73.72 | **74.15** |
| | ResNet-101 | 75.53 | 75.79 | **76.27** | * |
| Cityscapes | ResNet-50 | 73.17 | 75.52 | 76.09 | **76.52** |
| | ResNet-101 | 77.04 | 77.02 | **77.63** | * |

Table 1: **Bayesian inference for semantic segmentation.** The performance (Mean-IoU) of SGLD with learned priors (both supervised and SSL) and a non-learned prior over ResNet-50 and ResNet-101 parameters for segmentation tasks. *SimCLR ResNet-101 checkpoints have not been released.

## 5   Discussion

Our work reveals several new key insights about deep transfer learning:

- *Modifying the loss surface on the downstream task through informative priors leads to significant performance gains, with and without Bayesian inference.*
- *Bayesian inference provides a particular performance boost with the informative priors.*
- *Informative priors lead to more data-efficient performance on the downstream task.*
- *The success of this approach depends on capturing key directions in the loss surface of the source task, which we represent through a low-rank plus diagonal covariance matrix.*
- *Standard transfer learning can be significantly miscalibrated, even providing worse likelihood than Bayesian methods from scratch on the downstream task.*
- *Priors learned via self-supervised pre-training transfer better than those learning via supervised learning.*

In short, pre-training your loss with care provides an easy drop-in replacement for conventional transfer learning that relies on initialization.

