## Acknowledgements

We thank Pavel Izmailov, Polina Kirichenko, and Wesley Maddox for helpful discussions. This research is supported by NSF CAREER IIS-2145492, NSF I-DISRE 193471, NIH R01DA048764-01A1, NSF IIS-1910266, NSF 1922658 NRT-HDR: FUTURE Foundations, Translation, and Responsibility for Data Science, Meta Core Data Science, Google AI Research, BigHat Biosciences, Capital One, and an Amazon Research Award.

## References

[1] Rishi Bommasani, Drew A Hudson, Ehsan Adeli, Russ Altman, Simran Arora, Sydney von Arx, Michael S Bernstein, Jeannette Bohg, Antoine Bosselut, Emma Brunskill, et al. On the opportunities and risks of foundation models. *arXiv preprint arXiv:2108.07258*, 2021.

[2] Rohitash Chandra and Arpit Kapoor. Bayesian neural multi-source transfer learning. *Neurocomputing*, 378:54–64, 2020.

[3] Liang-Chieh Chen, George Papandreou, Florian Schroff, and Hartwig Adam. Rethinking atrous convolution for semantic image segmentation. *arXiv preprint arXiv:1706.05587*, 2017.

[4] Liang-Chieh Chen, Yukun Zhu, George Papandreou, Florian Schroff, and Hartwig Adam. Encoder-decoder with atrous separable convolution for semantic image segmentation. In *Proceedings of the European conference on computer vision (ECCV)*, pages 801–818, 2018.

[5] Tianqi Chen, Emily Fox, and Carlos Guestrin. Stochastic gradient hamiltonian monte carlo. In *International conference on machine learning*, pages 1683–1691. PMLR, 2014.

[6] Ting Chen, Simon Kornblith, Mohammad Norouzi, and Geoffrey Hinton. A simple framework for contrastive learning of visual representations. In *International conference on machine learning*, pages 1597–1607. PMLR, 2020.

[7] Marius Cordts, Mohamed Omran, Sebastian Ramos, Timo Rehfeld, Markus Enzweiler, Rodrigo Benenson, Uwe Franke, Stefan Roth, and Bernt Schiele. The cityscapes dataset for semantic urban scene understanding. In *Proceedings of the IEEE conference on computer vision and pattern recognition*, pages 3213–3223, 2016.

[8] Zihang Dai, Hanxiao Liu, Quoc V Le, and Mingxing Tan. Coatnet: Marrying convolution and attention for all data sizes. *arXiv preprint arXiv:2106.04803*, 2021.

[9] Jia Deng, Wei Dong, Richard Socher, Li-Jia Li, Kai Li, and Li Fei-Fei. Imagenet: A large-scale hierarchical image database. In *2009 IEEE conference on computer vision and pattern recognition*, pages 248–255. Ieee, 2009.

[10] Jacob Devlin, Ming-Wei Chang, Kenton Lee, and Kristina Toutanova. Bert: Pre-training of deep bidirectional transformers for language understanding. *arXiv preprint arXiv:1810.04805*, 2018.

[11] Kien Do, Truyen Tran, and Svetha Venkatesh. Semi-supervised learning with variational bayesian inference and maximum uncertainty regularization. *arXiv preprint arXiv:2012.01793*, 2020.

[12] Alexey Dosovitskiy, Lucas Beyer, Alexander Kolesnikov, Dirk Weissenborn, Xiaohua Zhai, Thomas Unterthiner, Mostafa Dehghani, Matthias Minderer, Georg Heigold, Sylvain Gelly, et al. An image is worth 16x16 words: Transformers for image recognition at scale. *arXiv preprint arXiv:2010.11929*, 2020.

[13] Sayna Ebrahimi, Mohamed Elhoseiny, Trevor Darrell, and Marcus Rohrbach. Uncertainty-guided continual learning with bayesian neural networks. *arXiv preprint arXiv:1906.02425*, 2019.

[14] M. Everingham, L. Van Gool, C. K. I. Williams, J. Winn, and A. Zisserman. The pascal visual object classes (voc) challenge. *International Journal of Computer Vision*, 88(2):303–338, June 2010.

[15] Vincent Fortuin, Adrià Garriga-Alonso, Florian Wenzel, Gunnar Rätsch, Richard Turner, Mark van der Wilk, and Laurence Aitchison. Bayesian neural network priors revisited. *arXiv preprint arXiv:2102.06571*, 2021.

[16] Yansong Gao, Rahul Ramesh, and Pratik Chaudhari. Deep reference priors: What is the best way to pretrain a model? *arXiv preprint arXiv:2202.00187*, 2022.

[17] Danijar Hafner, Dustin Tran, Timothy Lillicrap, Alex Irpan, and James Davidson. Noise contrastive priors for functional uncertainty. In *Uncertainty in Artificial Intelligence*, pages 905–914. PMLR, 2020.

[18] Kaiming He, Xiangyu Zhang, Shaoqing Ren, and Jian Sun. Deep residual learning for image recognition. In *Proceedings of the IEEE conference on computer vision and pattern recognition*, pages 770–778, 2016.

[19] Kaiming He, Haoqi Fan, Yuxin Wu, Saining Xie, and Ross Girshick. Momentum contrast for unsupervised visual representation learning. In *Proceedings of the IEEE/CVF Conference on Computer Vision and Pattern Recognition*, pages 9729–9738, 2020.

[20] Jeremy Howard and Sebastian Ruder. Universal language model fine-tuning for text classification. *arXiv preprint arXiv:1801.06146*, 2018.

[21] W Ronny Huang, Zeyad Emam, Micah Goldblum, Liam Fowl, Justin K Terry, Furong Huang, and Tom Goldstein. Understanding generalization through visualizations. *"I Can't Believe It's Not Better!" NeurIPS 2020 workshop*, 2020.

[22] Alexander Immer, Matthias Bauer, Vincent Fortuin, Gunnar Rätsch, and Khan Mohammad Emtiyaz. Scalable marginal likelihood estimation for model selection in deep learning. In *International Conference on Machine Learning*, pages 4563–4573. PMLR, 2021.

[23] Pavel Izmailov, Dmitrii Podoprikhin, Timur Garipov, Dmitry Vetrov, and Andrew Gordon Wilson. Averaging weights leads to wider optima and better generalization. *arXiv preprint arXiv:1803.05407*, 2018.

[24] Pavel Izmailov, Patrick Nicholson, Sanae Lotfi, and Andrew Gordon Wilson. Dangers of bayesian model averaging under covariate shift. *arXiv preprint arXiv:2106.11905*, 2021.

[25] Pavel Izmailov, Sharad Vikram, Matthew D Hoffman, and Andrew Gordon Wilson. What are bayesian neural network posteriors really like? *arXiv preprint arXiv:2104.14421*, 2021.

[26] Neal Jean, Sang Michael Xie, and Stefano Ermon. Semi-supervised deep kernel learning: Regression with unlabeled data by minimizing predictive variance. *Advances in Neural Information Processing Systems*, 31, 2018.

[27] Sanyam Kapoor, Theofanis Karaletsos, and Thang D Bui. Variational auto-regressive gaussian processes for continual learning. In *International Conference on Machine Learning*, pages 5290–5300. PMLR, 2021.

[28] Alireza Karbalayghareh, Xiaoning Qian, and Edward R Dougherty. Optimal bayesian transfer learning. *IEEE Transactions on Signal Processing*, 66(14):3724–3739, 2018.

[29] James Kirkpatrick, Razvan Pascanu, Neil Rabinowitz, Joel Veness, Guillaume Desjardins, Andrei A Rusu, Kieran Milan, John Quan, Tiago Ramalho, Agnieszka Grabska-Barwinska, et al. Overcoming catastrophic forgetting in neural networks. *Proceedings of the national academy of sciences*, 114(13):3521–3526, 2017.

[30] Alex Krizhevsky. Learning multiple layers of features from tiny images. Technical report, University of Toronto, 2009.

[31] Hao Li, Zheng Xu, Gavin Taylor, Christoph Studer, and Tom Goldstein. Visualizing the loss landscape of neural nets. *arXiv preprint arXiv:1712.09913*, 2017.

[32] Hao Li, Pratik Chaudhari, Hao Yang, Michael Lam, Avinash Ravichandran, Rahul Bhotika, and Stefano Soatto. Rethinking the hyperparameters for fine-tuning. In *International Conference on Learning Representations*, 2019.

[33] Wesley Maddox, Shuai Tang, Pablo Moreno, Andrew Gordon Wilson, and Andreas Damianou. Fast adaptation with linearized neural networks. In *International Conference on Artificial Intelligence and Statistics*, pages 2737–2745. PMLR, 2021.

[34] Wesley J Maddox, Pavel Izmailov, Timur Garipov, Dmitry P Vetrov, and Andrew Gordon Wilson. A simple baseline for bayesian uncertainty in deep learning. *Advances in Neural Information Processing Systems*, 32:13153–13164, 2019.

[35] Cuong V Nguyen, Yingzhen Li, Thang D Bui, and Richard E Turner. Variational continual learning. *arXiv preprint arXiv:1710.10628*, 2017.

[36] Maria-Elena Nilsback and Andrew Zisserman. Automated flower classification over a large number of classes. In *2008 Sixth Indian Conference on Computer Vision, Graphics & Image Processing*, pages 722–729. IEEE, 2008.

[37] Pingbo Pan, Siddharth Swaroop, Alexander Immer, Runa Eschenhagen, Richard Turner, and Mohammad Emtiyaz E Khan. Continual deep learning by functional regularisation of memorable past. *Advances in Neural Information Processing Systems*, 33:4453–4464, 2020.

[38] Omkar M Parkhi, Andrea Vedaldi, Andrew Zisserman, and CV Jawahar. Cats and dogs. In *2012 IEEE conference on computer vision and pattern recognition*, pages 3498–3505. IEEE, 2012.

[39] Massimiliano Patacchiola, Jack Turner, Elliot J Crowley, Michael O'Boyle, and Amos J Storkey. Bayesian meta-learning for the few-shot setting via deep kernels. *Advances in Neural Information Processing Systems*, 33:16108–16118, 2020.

[40] Valerio Perrone, Huibin Shen, Matthias W Seeger, Cedric Archambeau, and Rodolphe Jenatton. Learning search spaces for bayesian optimization: Another view of hyperparameter transfer learning. *Advances in Neural Information Processing Systems*, 32, 2019.

[41] Naresh Balaji Ravichandran, Anders Lansner, and Pawel Herman. Semi-supervised learning with bayesian confidence propagation neural network. *arXiv preprint arXiv:2106.15546*, 2021.

[42] Benjamin Recht, Rebecca Roelofs, Ludwig Schmidt, and Vaishaal Shankar. Do cifar-10 classifiers generalize to cifar-10? *arXiv preprint arXiv:1806.00451*, 2018.

[43] Shaoqing Ren, Kaiming He, Ross Girshick, and Jian Sun. Faster r-cnn: Towards real-time object detection with region proposal networks. *Advances in neural information processing systems*, 28:91–99, 2015.

[44] Daniel M Roy and Leslie Pack Kaelbling. Efficient bayesian task-level transfer learning. In *IJCAI*, volume 7, pages 2599–2604, 2007.

[45] Sebastian Ruder and Barbara Plank. Learning to select data for transfer learning with bayesian optimization. *arXiv preprint arXiv:1707.05246*, 2017.

[46] Tim GJ Rudner, Freddie Bickford Smith, Qixuan Feng, Yee Whye Teh, and Yarin Gal. Continual learning via function-space variational inference. In *ICML Workshop on Theory and Foundations of Continual Learning*, 2021.

[47] Dominik Schnaus. *Progressive Bayesian Neural Networks*. PhD thesis, Technical University of Munich, 2021.

[48] Ravid Shwartz-Ziv, Randall Balestriero, and Yann LeCun. What do we maximize in self-supervised learning? *arXiv preprint arXiv:2207.10081*, 2022.

[49] Hanna Tseran, Mohammad Emtiyaz Khan, Tatsuya Harada, and Thang D Bui. Natural variational continual learning. In *Continual Learning Workshop@ NeurIPS*, volume 2, 2018.

[50] Max Welling and Yee W Teh. Bayesian learning via stochastic gradient langevin dynamics. In *Proceedings of the 28th international conference on machine learning (ICML-11)*, pages 681–688. Citeseer, 2011.

[51] Andrew Gordon Wilson and Pavel Izmailov. Bayesian deep learning and a probabilistic perspective of generalization. *arXiv preprint arXiv:2002.08791*, 2020.

[52] Junyu Xuan, Jie Lu, and Guangquan Zhang. Bayesian transfer learning: An overview of probabilistic graphical models for transfer learning. *arXiv preprint arXiv:2109.13233*, 2021.

[53] Ruqi Zhang, Chunyuan Li, Jianyi Zhang, Changyou Chen, and Andrew Gordon Wilson. Cyclical stochastic gradient mcmc for bayesian deep learning. *arXiv preprint arXiv:1902.03932*, 2019.


| (a) Learning a prior | (b) Prior rescaling | (c) Bayesian inference |

Figure 5: **Bayesian transfer learning pipeline.** We (a) learn a prior by fitting a probability distribution over feature extractor parameters to a pre-training posterior mode, (b) rescale the prior for a downstream task, and (c) use the prior for Bayesian inference on downstream tasks.

## A    Computational Considerations

While our method requires a learned prior and a re-scaling coefficient, it actually adds very little additional cost on top of standard fine-tuning routines. We now evaluate the costs of each of the three stages of our pipeline: (i) inferring the posterior on the source task; (ii) re-scaling the posterior to become an informative prior for the downstream task; (iii) using the informative prior in the downstream task.

For (i), we note that we used SWAG because it has already been well-established as a tractable, well-studied, and simple way to infer a non-diagonal posterior, and it can be used starting from a pre-trained checkpoint [34]. We apply SWAG to a pre-trained checkpoint. We then do 50 iterations of burn-in and 10 cycles of 200 iterations each for a total of 2050 iterations with a batch size of 128. In total, inferring the posterior costs only around a quarter of an epoch of training on ImageNet, and the computations in this step can be entirely avoided by users releasing pre-trained SWAG posteriors, as we do for all of the tasks in our paper.

For (ii), we only need to tune a single re-scaling hyperparameter. When we perform MAP optimization with our informative priors, tuning this scalar only adds $\frac{1}{7}$ of the total runtime of transfer learning (including tuning other hyperparameters such as learning rate). For example, full hyperparameter tuning (including the re-scaling coefficient) and running MAP on the full Oxford Flowers-102 dataset takes 20 hours in total, whereas hyperparameter tuning and running traditional SGD-based transfer learning takes 17 hours in total on a machine with a single NVIDIA A100 GPU, which is a comparable amount of hyperparameter tuning to other works and significantly less than others [32, 6]. We also note that our tuning is done with a simple validation grid-search, exactly the same way as any other hyperparameter tuning in standard transfer learning, and does not require any special expertise.

For (iii), we offer two options, (a) MAP optimization of the downstream posterior using the informative prior, and (b) full Bayesian inference with SGLD or SGHMC using the downstream posterior. (a) costs the same as standard transfer learning both at train and test time, and we show it works better than standard transfer learning. (b) Bayesian inference with our informative priors bears the same cost as deep ensembles or as Bayesian inference with non-learning priors, which we compare to in our work. A single chain of SGLD incurs the same cost as a single run of SGD, since the gradient of the log prior density can be computed without automatic differentiation in the same fashion as weight-decay, which we are replacing. We also compare multiple chains of SGLD with deep ensembles, which require the same cost, and show that Bayesian inference with our learned prior works significantly better.

## B    Classification

### B.1    Implementation Details

Based on the evaluation protocols in the papers introducing the datasets, we report the top-one accuracy for CIFAR-10 and CIFAR-100 and mean per-class accuracy for Oxford-IIIT Pets and Oxford 102 Flowers. For Oxford 102 Flowers, we select hyperparameters based on the validation

sets specified by the dataset creators. While tuning hyperparameters, we hold out a subset of the training set for validation on the other datasets. After selecting the optimal hyperparameters from the validation set, we retrain the model using the selected parameters on both the training and validation images. Test results are reported.

We use ResNet-50 and ResNet-101 architectures [18] with supervised and SSL checkpoints [6, 48] pre-trained on the ImageNet 1k dataset [9]. We use the same hyperparameters as in Chen et al. [6].

**Learning the prior.** In order to learn the prior, we use the SWAG algorithm [34] with cyclic learning rate presented in Zhang et al. [53]. We use SGD with a Nesterov momentum parameter of 0.9 for optimization. We select our initial learning rate from a logarithmic grid between 0.005 and 0.5 and evaluate 10000 iterations for each cycle. Other hyperparameters are from Chen et al. [6], including data augmentation which comprises color augmentation, blurring, random crops, and horizontal flips.

**Downstream classification tasks.** We train for $30,000$ steps with a batch size of $128$ on CIFAR-10 and CIFAR-100, 16 for Oxford Flowers-102, and 32 for Oxford-IIIT Pets. We use SGD and SGHMC with momentum parameter of $0.9$. During fine-tuning, we perform random crops with resizing to $224 \times 224$ and horizontal flips. At test time, we resize the images to $256$ pixels along the shorter side and produce a $224 \times 224$ center crop. In our study, we select the learning rate and weight decay from a grid of 7 logarithmically spaced learning rates between $0.0001$ and $0.1$, and 7 logarithmically spaced weight decay values between $1e^{-6}$ and $1e^{-2}$, as well as without weight decay. These weight decay values are divided by the learning rate. For the SGHMC optimizer, we select the temperature from a logarithmic grid of $8$ values between $1e^0$ and $1e^{-8}$, and we use predictions from 5 different chains for the final evaluation.

## B.2 Supervised Pre-Training

As mentioned above, we conduct additional experiments with priors learned on labeled data. Figures 7 and 6 contain downstream task performance comparisons for Resnet-50 and Resnet-101 models with priors learned starting with `torchvision` checkpoints pre-trained on ImageNet 1k. These figures compare our method to SGD transfer learning with pre-trained initializations and Bayesian inference with non-learned Gaussian prior. Across both backbones, our method outperforms all baselines for nearly every number of train samples and for all four datasets, where the Bayesian inference model with a non-learned prior is always worse. (For full results see tables 2, 3, 4 and 5.

| Model/# samples | 10 | 100 | 1000 | 10000 | 50000 |
|---|---|---|---|---|---|
| BNN Learned Prior | 75.7 | 46.8 | 24.9 | 9.2 | 4.3 |
| BNN Non-learned Prior | 86.5 | 68.1 | 44.9 | 11.6 | 4.8 |
| SGD Ensemble | 78.7 | 57.2 | 31.6 | 10.9 | 5.2 |
| SGD Transfer Init | 81.9 | 62.1 | 36.5 | 11.6 | 4.4 |

Table 2: CIFAR-10 test error with `torchvision` Resnet-50 prior/initialization.

| Model/# samples | 10 | 100 | 1000 | 10000 | 50000 |
|---|---|---|---|---|---|
| BNN Learned Prior | 86.8 | 64.9 | 38.8 | 26.1 | 17.2 |
| BNN Non-learned Prior | 97.0 | 83.6 | 50.8 | 30.1 | 19.2 |
| SGD Ensemble | 93.3 | 74.6 | 43.1 | 26.4 | 17.4 |
| SGD Transfer Init | 94.9 | 71.8 | 42.7 | 26.9 | 17.8 |

Table 3: CIFAR-100 test error with `torchvision` Resnet-50 prior/initialization.

## B.3 Self-Supervised Pre-Training

In this section, we present additional data from Figures 3 and 4 found in the main body. Tables 10, 11, 12 and 13 correspond to Figure 3, while Table 14 corresponds to Figure 4.

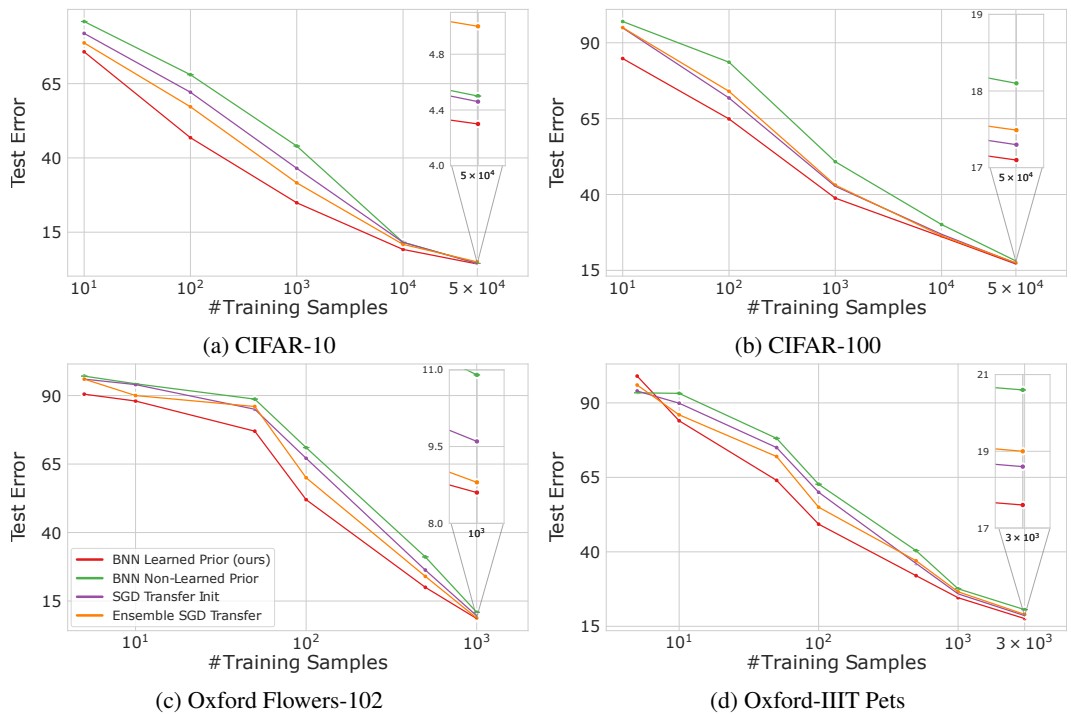

(a) CIFAR-10

(b) CIFAR-100

(c) Oxford Flowers-102

(d) Oxford-IIIT Pets

Figure 6: **Performance comparison.** The test error of a ResNet-50 BNN equipped with supervised pre-trained prior (red) is consistently lower than that of standard SGD-based transfer learning from the same pre-trained checkpoint (purple), an equivalently sized ensemble of SGD-based transfer learning models (orange), or BNN with a mean zero isotropic Gaussian prior (green). The x-axis denotes the number of downstream training samples. Axes on a logarithmic scale.

| Model/# samples | 5 | 10 | 50 | 100 | 500 | 1000 |
|---|---|---|---|---|---|---|
| BNN Learned Prior | 90.5 | 88.0 | 77.2 | 52.8 | 20.4 | 8.5 |
| BNN Non-learned Prior | 97.1 | 94.3 | 88.7 | 71.0 | 35.0 | 10.7 |
| SGD Ensemble | 96.2 | 90.1 | 86.1 | 60.0 | 24.0 | 8.9 |
| SGD Transfer Init | 96.6 | 94.3 | 85.4 | 68.0 | 26.3 | 9.6 |

Table 4: Oxford-Flowers-102 test error with `torchvision` Resnet-50 prior/initialization.

## B.4 Evaluating uncertainty

In Figure 8, we present the test negative log likelihood (NLL) for both the Bayesian methods and SGD-based methods for 4 datasets (CIFAR-10, CIFAR-100, Oxford Flowers-102 and Oxford-IIIT Pets). In general, we can see a similar trend for all the datasets where Bayesian transfer learning outperforms all other methods. In Figure 9, we present the reliability diagrams for CIFAR-10 and CIFAR-100 for ResNet-18. A perfectly calibrated network would express no difference between accuracy and confidence, represented by a dashed black line. Under-confident predictions are those below this line, whereas overconfident predictions are those above. We can see that Bayesian inference with learned priors are the best calibrated among the methods we consider. The error bars are computed on 5 runs, and the radius is one standard error.

## C   Semantic Segmentation

### C.1   Implementation Details

For our semantic segmentation evaluations, we use the DeepLabv3+ [4] model with ResNet-50 and ResNet-101 backbone architectures. We train our models using Pascal VOC 2012 and Cityscapes trainsets and evaluate on their *val* sets. We utilize the same hyperparameters employed in Chen et al.

| Model/# samples | 5 | 10 | 50 | 100 | 500 | 1000 | 3000 |
|---|---|---|---|---|---|---|---|
| BNN Learned Prior | 98.2 | 84.0 | 64.7 | 49.3 | 32.0 | 24.6 | 17.6 |
| BNN Non-learned Prior | 93.4 | 93.2 | 78.1 | 62.7 | 45.4 | 38.6 | 25.2 |
| SGD Ensemble | 96.6 | 86.3 | 72.1 | 55.0 | 37.9 | 26.6 | 19.3 |
| SGD Transfer Init | 94.5 | 89.9 | 75.4 | 61.1 | 36.1 | 25.9 | 18.6 |

Table 5: Oxford-IIIT Pets test error with `torchvision` Resnet-50 prior/initialization.

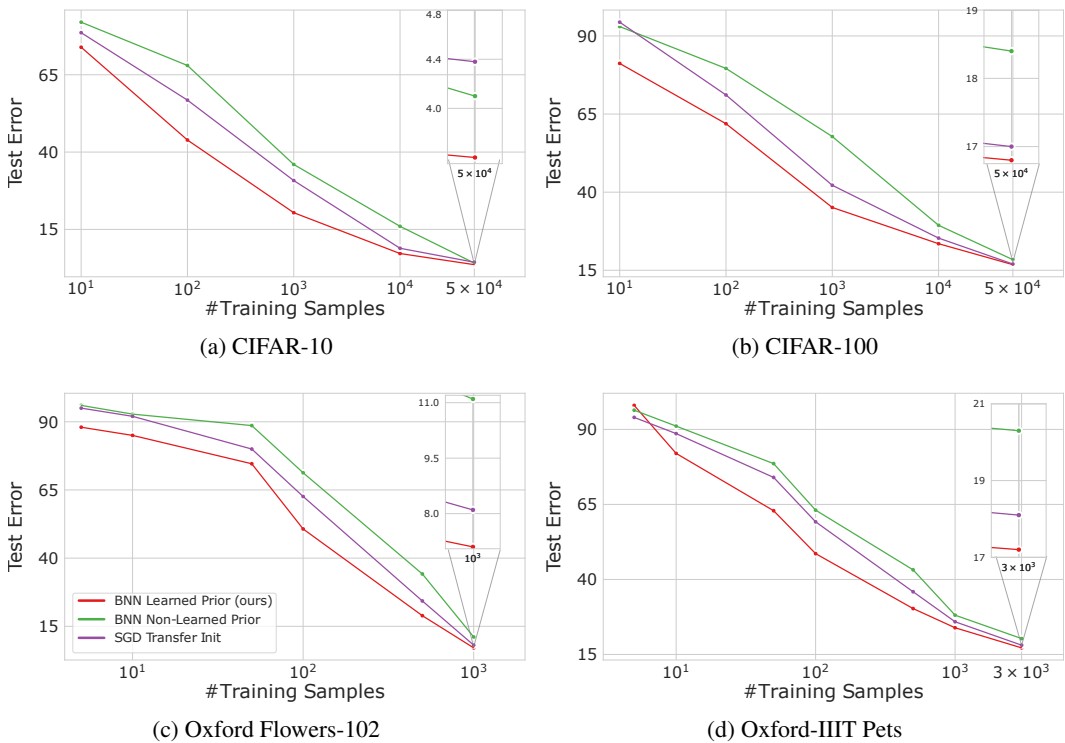

(a) CIFAR-10

(b) CIFAR-100

(c) Oxford Flowers-102

(d) Oxford-IIIT Pets

Figure 7: **Performance comparison.** The test error of a ResNet-101 BNN equipped with supervised pre-trained prior (red) is consistently lower than that of standard SGD-based transfer learning from the same pre-trained checkpoint (purple), or BNN with a mean zero isotropic Gaussian prior (green). The x-axis denotes the number of downstream training samples. Axes on a logarithmic scale.

[4] with a few modifications: our learning rate schedule is the *poly* policy [3] with initial value $0.01$ for Pascal VOC 2012 and $0.1$ for Cityscapes, our crop size is $513 \times 513$ for Pascal VOC 2012 and $768 \times 768$ for Cityscapes. Our *output stride* is 16 for both training and evaluating, and we perform random scale data augmentation during training. In our SGD and SGLD optimization, we employ Nesterov momentum, and we set the momentum coefficient to $0.9$. We use batches of 16 images and weight decay with a coefficient of $10^{-4}$. In addition, for the SGLD optimizer, the temperature is set to $2 \times 10^{-8}$. We train all models for 30k iterations.

## C.2   MAP Estimation

We also compare MAP estimates (SGD) on the loss induced by our learned priors to the baseline with only weight decay. Table 15 and 16 contain the results using ResNet-50 and ResNet-101 backbones, respectively. On both architectures and both datasets, learned priors boost performance on these experiments without using Bayesian inference at all.

## D   Alternative Methods for Re-Scaling the Prior

As discussed previously, we tune a coefficient to re-scale our learned prior by maximizing the validation log-likelihood over a grid of coefficient values. Our objective in this section is to examine

| Model/# samples | 10 | 100 | 1000 | 10000 | 50000 |
|---|---|---|---|---|---|
| BNN Learned Prior | 73.9 | 43.9 | 20.4 | 7.2 | 3.7 |
| BNN Non-learned Prior | 82.1 | 68.7 | 36.3 | 16.4 | 4.1 |
| SGD Transfer Init | 78.6 | 56.8 | 30.8 | 8.9 | 4.3 |

Table 6: CIFAR-10 test error with `torchvision` Resnet-101 prior/initialization.

| Model/# samples | 10 | 100 | 1000 | 10000 | 50000 |
|---|---|---|---|---|---|
| BNN Learned Prior | 81.2 | 61.9 | 35.1 | 23.5 | 16.8 |
| BNN Non-learned Prior | 93.5 | 79.6 | 58.8 | 32.0 | 20.2 |
| SGD Transfer Init | 94.4 | 71.1 | 44.2 | 25.9 | 17.0 |

Table 7: CIFAR-100 test error with `torchvision` Resnet-101 prior/initialization.

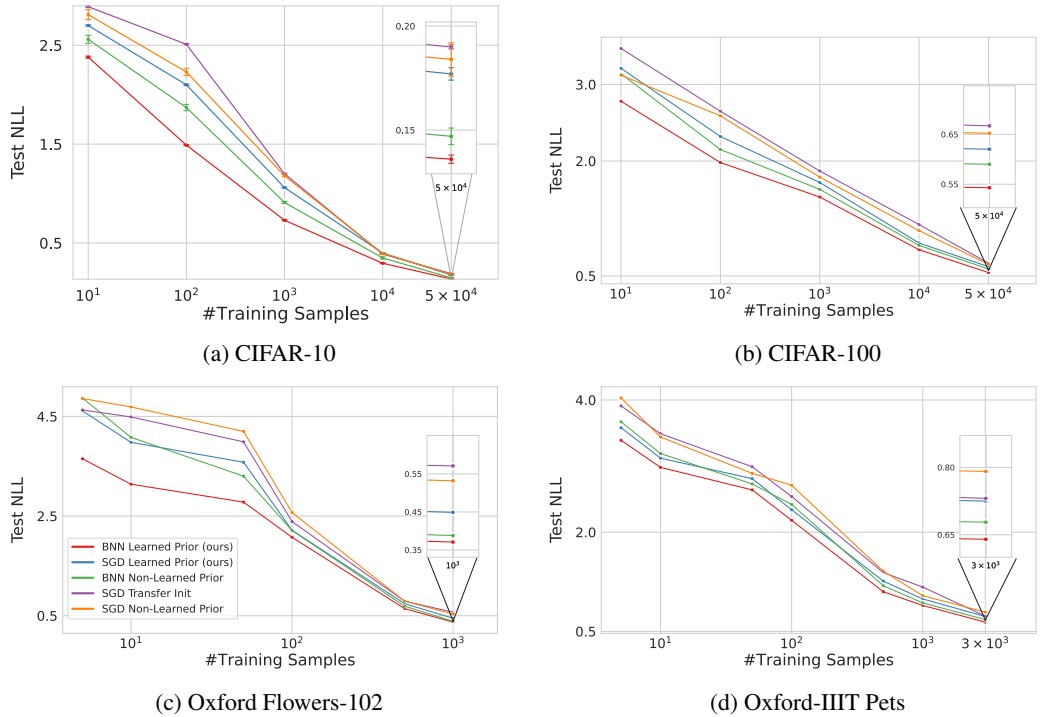

(a) CIFAR-10

(b) CIFAR-100

(c) Oxford Flowers-102

(d) Oxford-IIIT Pets

Figure 8: **Predictive likelihood.** The horizontal axis denotes the number of downstream CIFAR-10 training samples on a logarithmic scale. All experiments performed with ResNet-50 backbone.

three alternative methods for tuning coefficients to re-scale the prior. First, we consider maximizing the Laplace approximation to the marginal likelihood, which does not require holding out validation data [22]. Marginal likelihood maximization can be performed online together with MAP estimation in deep learning. Specifically, we try tuning a single coefficient and also per-layer coefficients for re-scaling the prior's covariance matrix in this fashion. A third alternative is to instead tune two coefficients: one for the diagonal component of the covariance matrix, and one for the low-rank component. We use a grid-search in this case to tune the two coefficients. Table 17 contains the results for these three different methods as well as the method advocated in the main body of our draft using the SimCLR Resnet-50 backbone and SGHMC for downstream fine-tuning. We observe that on most datasets, tuning a single covariance re-scaling hyperparameter using marginal likelihood does not improve the results reported in the paper. Using different coefficients for each layer results in slight improvements on some datasets. On CIFAR-10, Oxford Flowers-102, and Oxford-IIIT Pets, test errors decreased by $0.03\% - 0.08\%$. Finally, on most datasets, re-scaling the diagonal and low-rank components of the covariance matrix independently results in slightly better performance. For example, the test errors on CIFAR-100, Oxford Flowers-102, and Oxford-IIIT Pets decrease

| Model/# samples | 5 | 10 | 50 | 100 | 500 | 1000 |
|---|---|---|---|---|---|---|
| BNN Learned Prior | 88.0 | 81.0 | 74.6 | 50.7 | 18.9 | 7.1 |
| BNN Non-learned Prior | 96.0 | 92.8 | 88.6 | 75.1 | 41.1 | 21.7 |
| SGD Transfer Init | 95.0 | 90.2 | 80.8 | 59.2 | 22.3 | 8.6 |

Table 8: Oxford-Flowers-102 test error with `torchvision` Resnet-101 prior/initialization.

| Model/# samples | 5 | 10 | 50 | 100 | 500 | 1000 | 3000 |
|---|---|---|---|---|---|---|---|
| BNN Learned Prior | 96.1 | 82.4 | 62.9 | 48.6 | 30.3 | 23.9 | 17.2 |
| BNN Non-learned Prior | 96.4 | 91.1 | 78.6 | 67.7 | 56.9 | 38.2 | 26.2 |
| SGD Transfer Init | 94.0 | 88.6 | 74.3 | 59.2 | 35.5 | 25.0 | 18.1 |

Table 9: Oxford-IIIT Pets test error with `torchvision` Resnet-101 prior/initialization.

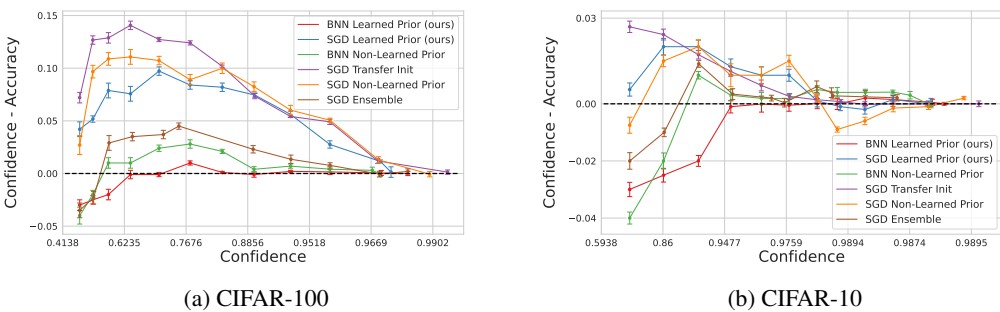

(a) CIFAR-100                                    (b) CIFAR-10

Figure 9: **Reliability diagrams** - Bayesian transfer learning is capable of significantly improving calibration over standard transfer training (SGD Transfer Init) as well as non-learned prior BNN. Error bars are computed over 5 runs.

by $0.03\% - 0.11\%$. We note that these alternative methods bear an additional computational and implementational cost.

# E   Limitations

*Limitations.* Bayesian model averaging provides particularly compelling results, especially in terms of calibration and data efficiency, but does incur some additional computational cost. On the other hand, our SGLD based sampling procedures outperform deep ensembles, which have comparable computational costs. Moreover, our informative priors with MAP optimization still provide a clear performance boost over standard transfer learning, with negligible overhead. In terms of applications, we have limited our considerations to vision settings, for a focused exposition.

*Directions for future research.* Pre-training has been particularly transformative in natural language processing, where informative priors could be used to great effect. More broadly studying loss-surface alignment between tasks could also be informative for understanding how to build models that provide good domain generalization and robustness to spurious correlations.

| Model/# samples | 10 | 100 | 1000 | 10000 | 50000 |
|---|---|---|---|---|---|
| BNN Learned Prior | $73.8 \pm 3.2$ | $43.1 \pm 3.1$ | $21.6 \pm 1.6$ | $7.8 \pm 0.5$ | $4.0 \pm 0.1$ |
| SGD Learned Prior | $75.9 \pm 2.1$ | $49.3 \pm 1.9$ | $26.8 \pm 1.1$ | $8.9 \pm 0.3$ | $4.2 \pm 0.1$ |
| BNN LP Single-Chain | $74.0 \pm 2.7$ | $48.4 \pm 2.8$ | $25.4 \pm 1.3$ | $8.3 \pm 0.5$ | $4.2 \pm 0.2$ |
| BNN LP Laplace | $86.9 \pm 3.1$ | $66.4 \pm 2.2$ | $37.1 \pm 2.1$ | $11.2 \pm 0.6$ | $4.6 \pm 0.2$ |
| BNN Non-learned Prior | $86.1 \pm 3.9$ | $68.0 \pm 3.8$ | $44.9 \pm 1.7$ | $11.6 \pm 0.6$ | $4.4 \pm 0.1$ |
| SGD Ensemble | $81.3 \pm 2.2$ | $57.0 \pm 3.1$ | $32.3 \pm 1.4$ | $9.4 \pm 0.5$ | $4.6 \pm 0.1$ |
| SGD Transfer Init | $83.6 \pm 1.8$ | $61.0 \pm 2.1$ | $36.8 \pm 1.0$ | $10.9 \pm 0.4$ | $4.4 \pm 0.1$ |

Table 10: CIFAR-10 test error with SimCLR (SSL) Resnet-50 prior/initialization.

| Model/# samples | 10 | 100 | 1000 | 10000 | 50000 |
|---|---|---|---|---|---|
| BNN Learned Prior | $81.9 \pm 3.7$ | $63.8 \pm 2.4$ | $37.3 \pm 1.2$ | $24.2 \pm 0.7$ | $16.4 \pm 0.4$ |
| SGD Learned Prior | $90.9 \pm 2.8$ | $69.7 \pm 1.9$ | $41.1 \pm 1.1$ | $26.3 \pm 0.5$ | $17.3 \pm 0.4$ |
| BNN LP Laplace | $90.1 \pm 3.1$ | $73.6 \pm 2.5$ | $42.8 \pm 1.9$ | $27.9 \pm 1.2$ | $18.1 \pm 0.8$ |
| BNN LP Single-Chain | $84.4 \pm 3.8$ | $69.6 \pm 2.1$ | $38.2 \pm 2.$ | $25.1 \pm 0.7$ | $16.8 \pm 0.5$ |
| BNN Non-learned Prior | $97.0 \pm 3.9$ | $84.0 \pm 2.8$ | $50.8 \pm 1.8$ | $28.9 \pm 0.8$ | $18.8 \pm 0.6$ |
| SGD Ensemble | $89.2 \pm 2.0$ | $70.9 \pm 1.7$ | $44.0 \pm 1.2$ | $26.6 \pm 0.7$ | $17.2 \pm 0.4$ |
| SGD Transfer Init | $93.0 \pm 2.2$ | $73.2 \pm 1.8$ | $45.5 \pm 0.9$ | $27.8 \pm 0.6$ | $17.9 \pm 0.3$ |

Table 11: CIFAR-100 test error with SimCLR (SSL) Resnet-50 prior/initialization.

| Model/# samples | 5 | 10 | 50 | 100 | 500 | 1000 |
|---|---|---|---|---|---|---|
| BNN Learned Prior | $89.8 \pm 3.1$ | $80.3 \pm 1.7$ | $67.3 \pm 1.8$ | $48.9 \pm 1.4$ | $19.7 \pm 0.7$ | $7.8 \pm 0.2$ |
| SGD Learned Prior | $90.2 \pm 2.0$ | $81.0 \pm 1.3$ | $67.0 \pm 1.1$ | $49.9 \pm 1.1$ | $20.6 \pm 0.6$ | $8.2 \pm 0.1$ |
| BNN LP Laplace | $97.4 \pm 2.3$ | $93.1 \pm 2.1$ | $89.9 \pm 1.4$ | $69.8 \pm 1.4$ | $32.1 \pm 0.8$ | $11.9 \pm 0.2$ |
| BNN LP Single-Chain | $90.3 \pm 2.8$ | $80.8 \pm 1.8$ | $67.4 \pm 1.1$ | $49.5 \pm 1.7$ | $20.1 \pm 0.9$ | $8.1 \pm 0.2$ |
| BNN Non-learned Prior | $97.1 \pm 2.7$ | $94.3 \pm 2.0$ | $88.7 \pm 2.1$ | $71.0 \pm 2.0$ | $35.0 \pm 0.9$ | $10.9 \pm 0.3$ |
| SGD Ensemble | $93.8 \pm 2.0$ | $91.2 \pm 1.6$ | $84.3 \pm 1.2$ | $56.8 \pm 1.0$ | $21.9 \pm 0.3$ | $8.5 \pm 0.2$ |
| SGD Transfer Init | $96.0 \pm 1.8$ | $93.0 \pm 1.8$ | $82.0 \pm 1.3$ | $66.0 \pm 1.1$ | $24.1 \pm 0.6$ | $8.7 \pm 0.1$ |

Table 12: Oxford-Flowers-102 test error with SimCLR (SSL) Resnet-50 prior/initialization.

| Model/# samples | 5 | 10 | 50 | 100 | 500 | 1000 | 3000 |
|---|---|---|---|---|---|---|---|
| BNN Learned Prior | $89.4 \pm 2.6$ | $80.9 \pm 1.9$ | $62.0 \pm 1.8$ | $47.3 \pm 1.6$ | $30.7 \pm 1.1$ | $21.4 \pm 0.9$ | $17.1 \pm 0.7$ |
| SGD Learned Prior | $91.7 \pm 1.9$ | $84.7 \pm 1.7$ | $66.1 \pm 1.9$ | $51.2 \pm 1.3$ | $34.1 \pm 0.8$ | $23.9 \pm 0.6$ | $18.2 \pm 0.6$ |
| BNN LP Laplace | $94.1 \pm 3.2$ | $91.1 \pm 2.7$ | $75.9 \pm 2.7$ | $60.8 \pm 1.7$ | $40.1 \pm 1.4$ | $26.8 \pm 1.4$ | $21.3 \pm 0.3$ |
| BNN LP Single-Chain | $91.0 \pm 2.7$ | $84.9 \pm 2.5$ | $65.2 \pm 2.9$ | $49.4 \pm 1.9$ | $31.9 \pm 1.8$ | $23.7 \pm 1.6$ | $18.1 \pm 0.6$ |
| BNN Non-learned Prior | $93.4 \pm 2.9$ | $93.2 \pm 2.1$ | $78.1 \pm 2.2$ | $62.7 \pm 2.1$ | $45.4 \pm 1.1$ | $38.6 \pm 1.1$ | $25.2 \pm 0.8$ |
| SGD Ensemble | $92.2 \pm 2.1$ | $88.1 \pm 1.4$ | $68.3 \pm 2.1$ | $55.7 \pm 1.5$ | $35.0 \pm 1.2$ | $24.5 \pm 1.0$ | $18.1 \pm 0.5$ |
| SGD Transfer Init | $94.9 \pm 2.1$ | $88.5 \pm 1.6$ | $72.0 \pm 1.3$ | $55.7 \pm 1.1$ | $36.3 \pm 1.0$ | $25.0 \pm 0.7$ | $18.9 \pm 0.3$ |

Table 13: Oxford-IIIT Pets test error with SimCLR (SSL) Resnet-50 prior/initialization.

| Model/# samples | 10 | 100 | 1000 | 10000 | 50000 |
|---|---|---|---|---|---|
| BNN Learned Prior | $81.8 \pm 2.0$ | $59.9 \pm 1.7$ | $43.0 \pm 1.3$ | $18.7 \pm 0.8$ | $8.8 \pm 0.4$ |
| SGD Learned Prior | $81.6 \pm 1.4$ | $63.9 \pm 1.5$ | $48.0 \pm 1.2$ | $20.1 \pm 0.8$ | $9.8 \pm 0.3$ |
| BNN Non-learned Prior | $86.9 \pm 2.3$ | $78.5 \pm 1.8$ | $59.0 \pm 1.3$ | $32.0 \pm 1.0$ | $11.3 \pm 0.4$ |
| SGD Transfer Init | $83.1 \pm 1.6$ | $69.0 \pm 1.3$ | $52.0 \pm 1.1$ | $21.3 \pm 0.6$ | $10.3 \pm 0.2$ |

Table 14: Classification error with self-supervised pre-learning - CIFAR-10.1

Table 15: **MAP estimation for semantic segmentation (ResNet-50 backbone).** Comparing the performance (Mean-IoU) of SGD with learned priors (both `torchvision` supervised and SimCLR SSL) to a non-learned prior (weight decay with `torchvision` initialization) over DeepLabv3+ backbone parameters for downstream segmentation with SGD rather than Bayesian inference. Evaluations are conducted on *val* sets.

| Dataset | Non-Learned Prior MAP | Supervised Prior MAP | SSL Prior MAP |
|---|---|---|---|
| PASCAL VOC 2012 | 73.17 | 73.27 | **73.48** |
| Cityscapes | 75.52 | 75.57 | **76.15** |

Table 16: **MAP estimation for semantic segmentation (ResNet-101 backbone).** Comparing the performance (Mean-IoU) of SGD with `torchvision` learned priors to a non-learned prior (weight decay with `torchvision` initialization) over DeepLabv3+ backbone parameters for downstream segmentation with SGD rather than Bayesian inference. Evaluations are conducted on *val* sets.

| Dataset | Non-Learned Prior MAP | Supervised Prior MAP |
|---|---|---|
| PASCAL VOC 2012 | 75.53 | **75.89** |
| Cityscapes | 77.04 | **77.27** |

Table 17: **Alternative methods for prior re-scaling (SimCLR ResNet-50 backbone)** - test error corresponding to different methods for re-scaling the prior.

| Dataset | CIFAR-100 | CIFAR-10 | Oxford Flowers-102 | Oxford-IIIT Pets |
|---|---|---|---|---|
| Single Coefficient - Grid Search | 17.14 | 4.02 | 7.85 | 17.12 |
| Single Coefficient - Marginal Likelihood | 17.13 | 4.02 | 7.87 | 17.06 |
| Per-Layer - Marginal Likelihood | 17.14 | 3.99 | 7.79 | 17.05 |
| Separate Scales for Low-Rank and Diagonal | 17.11 | 4.01 | 7.81 | 17.03 |