# OpenReview forum: "Pre-Train Your Loss: Easy Bayesian Transfer Learning with Informative Priors"
_NeurIPS.cc/2022/Conference — NeurIPS 2022 Accept_

### Official Review · Reviewer_hPGA · 2022-07-11

**Rating:** 6
**Confidence:** 4
**Soundness:** 4 excellent
**Presentation:** 4 excellent
**Contribution:** 3 good

**Summary:**

This paper proposes a Bayesian perspective of transfer learning in deep neural networks, whereby a re-scaled Bayesian parameter posterior from a source task is used as a pre-trained prior for a target task. The proposed procedure effectively reshapes the training objective of the target task to more faithfully reflect knowledge learnt from the source task. The authors find that modifying the loss surface of target tasks through informative priors significantly improves performance and calibration, especially with Bayesian inference.

**Questions:**

Since transformers (visual or otherwise) are the main subjects of transfer learning nowadays, it would make a lot of sense to evaluate them too. It is also known that transformers incur different loss landscapes than typical CNNs such as ResNets, it would be interesting to see how performance, robustness and calibration are effected by the proposed transfer learning policy.

**Limitations:**

See above.

**Strengths And Weaknesses:**

**Strengths**:

- Highly important and relevant topic for the research community;
- Clear and organized exposition;
- Well motivated arguments and experiments.

**Weaknesses**:
- Proposed approach can incur significant additional computational costs;
- May require expert knowledge to use effectively.

Overall I like this paper. I find the arguments and experiments convincing enough and believe it would be valuable for the research community. One concern is whether the proposed Bayesian inference based transfer learning pipeline would be of limited utility to regular practitioners, as expert knowledge may be required to get these systems working stably as intended. There is also the incurred computational cost which may deter uptake if the performance gain does not justify it.

---

> ### Author Response · Authors · 2022-08-02
> **Response to Reviewer hPGA**
>
> Thank you for your thoughtful and supportive feedback! We appreciate that you found the work well motivated and highly important, and the experiments convincing.
>
> We want to emphasize that our pipeline consists of three simple and well-established components, and thus is relatively easy to use, without requiring domain expertise. We believe the simplicity of the approach is actually a strength, especially in the context of the promising results. We highlight that the single extra hyperparameter we introduce can be tuned using a standard approach (a simple grid search on the validation set), and as per our general response (titled “General Response to Reviewers and AC”), incurs only a 1/7th additional cost over standard transfer learning (which includes tuning other hyperparameters such as learning rate). SWAG for posterior inference is also an extremely simple method which has been adopted in the community for several real-world applications and bears virtually no computational overhead if employed during pre-training as evidenced by [1]. While Bayesian inference on the downstream task can incur some overhead, we stress that our learned priors are highly effective with SGD fine-tuning as well (without Bayesian inference) as evidenced by Figure 3 and Tables 15 and 16 of the Appendix on both classification and segmentation, which incurs essentially no overhead.
>
> In general, Bayesian methods have made extraordinary practical progress in the last couple years for ConvNet and ResNet architectures on vision problems, but are almost entirely unexplored for transformers and NLP. As you point out, this would involve very different considerations, and is at this point an open question. We believe a Bayesian treatment of those areas could only properly be addressed in its own paper, which would not be specific to transfer learning. Our goal in this paper is to show how with a relatively simple and well-motivated approach, we can improve transfer learning pipelines with Bayesian principles, and we believe we provide relatively extensive thorough experiments, especially with the new updates to the paper we have made, detailed in the general response.
>
> Thank you again for your review and feedback. We hope you can consider raising your score in light of our response – we believe we are making many timely and significant contributions, outlined in the general response – and are happy to answer any further questions you might have.
>
>
> ### References:
> [1] Maddox, Wesley J., et al. "*A simple baseline for bayesian uncertainty in deep learning.*" Advances in Neural Information Processing Systems 32 (2019).

---

> > ### Author Response · Authors · 2022-08-06
> > **Response to Reviewer hPGA - Further engagement**
> >
> > Dear reviewer hPGA,
> >
> > Thank you again for your thoughtful review.
> >
> > Does our response address your questions?
> > We would appreciate the opportunity to engage further if needed.

---

### Official Review · Reviewer_3R6h · 2022-07-11

**Rating:** 5
**Confidence:** 2
**Soundness:** 3 good
**Presentation:** 4 excellent
**Contribution:** 2 fair

**Summary:**

This paper presents a Bayesian learning method for transfer learning on target task. Specifically, a posterior approximation method SWAG is used to estimate the posterior distribution of a supervised or self-supervised pre-trained model. This distribution is used as the prior for downstream learning. Experiments and ablation study demonstrate the effectiveness of learning a prior for transfer.

**Questions:**

Q1: Most of the evaluations in experiments focus on the performance of the learned prior when transferred to a new target task. I'm not sure if it is possible to use Bayesian model average can on source task itself, as it is not clear how good is the approximated posterior distribution. This study on the posterior approximation on the source task can probably show the effectiveness of estimation with close-form density as well as Bayesian learning.

Q2: The rescaling method proposed in section 3.2 is not well developed. Are there any related work for the proposed scaling method? Practically, from Fig. 2d, it seems that the scaling factor is very big. It seems that it is necessary to tune this scaling factor for every new target dataset. Is it possible to estimate this parameter without using holdout validation set?

Q3: Transfer learning results on target classification tasks are reported mainly using test error. A more common comparison is classification accuracies as error depends on the loss function. As seen from the tables the test error of several baseline methods are quite close. What are the classification accuracies for these results?



**Limitations:**

See weakness and questions.

**Strengths And Weaknesses:**

Strengths

1) The problem of learning a prior for transfer with Bayesian learning is well motivated.

2) The proposed method with three steps is reasonable.

3) Experimental results of the proposed method on semantic segmentation are quite strong.

Weaknesses

1) This paper employs methods of posterior approximation and Bayesian learning and incorporates them into Bayesian transfer by learning a prior from source tasks. These existing methods have been well developed so the overall technical contribution is incremental upon these.

2) The method introduced in section 3.2 looks a bit arbitrary. In order to slightly smooth the prior distribution, this paper proposes to multiply a single scaling factor to the covariance matrix. This does not seem to be a principled way to adapt a prior distribution to a target task. Are there any other possible methods of adjusting the covariance matrix for transfer? The holdout validation method used to select this scaling factor also neglects other data information that is potentially useful for determining the scaling factor.

3) The background review on Bayesian transfer learning seems not very thorough, which makes the overall contribution of the proposed transfer learning with Bayesian learning unclear. The only relevant review on this direction is about continual learning.

4) The proposed method will introduce additional computational burden for transfer learning.

---

> ### Author Response · Authors · 2022-08-02
> **Response to Reviewer 3R6h - Part 1**
>
> Thank you for your time and feedback! We note that in addition to this response post, we’ve also made a separate general post, with several clarifications and new results inspired by your comments, including a runtime analysis, a study of the scaling hyperparameter, and using a Laplace posterior from the source task, as an alternative to SWAG. We respond to each of your points below.
>
> **W1 (“...employs methods of posterior approximation and Bayesian learning…”):** It is true that our pipeline uses a number of simple existing components in order to tackle a distinct problem, transfer learning; however, combining them properly, e.g. choosing the right form of the covariance and rescaling the SWAG posterior, was essential for making our pipeline work so well.  Moreover, we believe the fact that such a simple pipeline can achieve such strong performance on hard downstream tasks like semantic segmentation is in fact a strength of the approach rather than a weakness, and the method’s simplicity, as well as the fact the various components are well-studied, will aid in adoption.
>
> **W2 (“...Are there any other possible methods of adjusting the covariance matrix for transfer?...”):**  We originally chose to tune a single re-scaling hyperparameter on a hold-out set for the sake of simplicity and ease-of-use — it can be set in the same way as any other parameter in standard transfer learning, without incurring any significant overhead (about 1/7th the total cost of tuning standard hypers), and requires no specialized domain expertise. It also has a simple interpretation: since source and downstream tasks come from different distributions, we need to ensure the posterior on the source task is not over- or under-concentrated for use as a prior in the downstream task.
>
> However, prompted by your review, we have tested several alternatives.  We first tried using a Laplace approximation to the marginal likelihood [1] to tune our re-scaling hyperparameter without having to hold out a validation set.  Scaling a single covariance parameter using marginal likelihood didn’t improve the results reported in the paper on most of the datasets.  Inspired by [3], we then tried using the same marginal likelihood approximation to tune individual scaling coefficients for each layer. This approach achieves slightly better results on some datasets.  For example, the test errors on CIFAR-10, Oxford Flowers-102, and Oxford-IIIT Pets decreased by 0.03%-0.08%. .  Finally, we tried tuning separate re-scaling coefficients for the low-rank and diagonal components of the covariance matrix. This method requires tuning an additional hyper-parameter but achieves slightly better performance on most datasets. For example, the test error on CIFAR-100, Oxford Flowers-102 and Oxford-IIIT Pets were lower by 0.03%-0.11%.
>
> As a result, we conclude that tuning a single rescaling coefficient is a simple and effective way to rescale the parameter, but some alternatives might yield slight improvements at the expense of compute and implementation cost. We have updated the draft of our paper to include these results, as well as a detailed discussion in our general response, and we appreciate your suggestion.
>
> ***W3 (“...background review on Bayesian transfer learning seems not very thorough…”):*** We note that we did have a discussion of several related areas beyond continual learning, such as leveraging auxiliary data with Bayesian methods via probabilistic graphical models, and semi-supervised deep kernel learning. However, prompted by your feedback, we have now extended our literature review regarding Bayesian transfer learning in the updated draft and cited additional sources on this topic, see Section 2 of our updated draft. If you have specific references which you think are important, please let us know, and we’ll add them.
>
> ***W4 (“...additional computational burden for transfer learning…”):***  We agree with you that learning a prior and a re-scaling coefficient increases the compute overhead for transfer learning, and we have added a more detailed discussion in our updated draft.  We do want to stress that learning the prior requires only a quarter of an epoch of SGD on ImageNet, and the re-scaling coefficient is only a single hyperparameter tuned in the same exact fashion as other hyperparameters typically tuned in transfer learning routines. Tuning the re-scaling hyperparameter only adds 1/7 total cost to SGD-based downstream fine-tuning and ⅛ total cost to downstream Bayesian inference considering the other hyperparameters, such as learning rate, those routines already require tuning.  See the general comment to all reviewers for further details.  We have added a corresponding discussion to Appendix A, which we will move to the main body with the additional page allowed for the camera-ready version.

---

> > ### Author Response · Authors · 2022-08-02
> > **Response to Reviewer 3R6h - Part 2**
> >
> >
> > ***Q1 (“I'm not sure if it is possible to use Bayesian model average can on source task itself, as it is not clear how good is the approximated posterior distribution”):***  We have now tested our source task posteriors on the ImageNet test set. The top-1 accuracy of our learned model on the ImageNet dataset is 76.24% compared to 76.15% for the original torchvision checkpoints, constituting a small increase. We also point out that SWAG posteriors have been thoroughly evaluated by the community outside of their transferability in [3] and numerous others.
> >
> > ***Q2 (“The rescaling method proposed in section 3.2 is not well developed…It seems that it is necessary to tune this scaling factor for every new target dataset…”):***  See our comments above regarding W2, and the general response.  Prompted by your review, we have delved much deeper into how the covariance matrix is best re-scaled and run numerous new experiments, which we have included in our updated draft.  Additionally, we would like to clarify that the scaling factor must necessarily be target dataset dependent since different downstream loss functions will vary in their consistency with the upstream loss.  In other words, some downstream loss functions will have good solutions where the upstream posterior has a high concentration of mass and will require less rescaling, while other downstream tasks will require more rescaling to add mass to the regions of their loss surface where solutions consistent with the downstream task reside.  Also, note that in Figure 2d, we show that performance is stable with respect to the scaling value, in fact, often holding for scalars within several orders of magnitude. In our paper, we typically checked only 6 values in this range and used early stopping when the validation loss was significantly lower than the best previous run (on average, this happens after only half of the total epochs).
> >
> > ***Q3 (“...A more common comparison is classification accuracies as error depends on the loss function…”):***  We use the convention error = 1 - accuracy.  Thus, you can exactly compute accuracy numbers from our results, and we have included all raw experimental results in the Appendix.  To further clarify, test error does not depend on the loss function at all and presents exactly the same information as classification accuracy. Test error is a common metric in well-regarded works such as the AlexNet paper [4] and the ResNet paper [5]. Note we also provide test likelihood comparisons.
> >
> > Thank you very much for your review. We have put a significant effort into responding to your points. We hope you can consider raising your score in light of our response.  Please let us know if you have additional questions we can address.
> >
> > ### References:
> > [1] Immer, Alexander, et al. "*Scalable marginal likelihood estimation for model selection in deep learning.*" International Conference on Machine Learning. PMLR, 2021.
> >
> > [2] Mackay, David John Cameron. "*Bayesian methods for adaptive models*." California Institute of Technology, 1992.
> >
> > [3] Maddox, Wesley J., et al. "*A simple baseline for bayesian uncertainty in deep learning.*" Advances in Neural Information Processing Systems 32 (2019).
> >
> > [4] Krizhevsky, Alex, Ilya Sutskever, and Geoffrey E. Hinton. "*Imagenet classification with deep convolutional neural networks.*" Advances in neural information processing systems 25 (2012).
> >
> > [5] He, Kaiming, et al. "*Deep residual learning for image recognition.*" Proceedings of the IEEE conference on computer vision and pattern recognition, 2016.

---

> > > ### Author Response · Authors · 2022-08-06
> > > **Response to Reviewer 3R6h - Further engagement**
> > >
> > > Dear reviewer 3R6h,
> > >
> > > Thank you again for your thoughtful review.
> > >
> > > Does our response address your questions? We would appreciate the opportunity to engage further if needed.

---

> > > ### Comment · Reviewer_3R6h · 2022-08-08
> > > **Thank you**
> > >
> > > Thank you for the detailed response to my earlier questions. The response clarifies a number of my questions. The extra experiments are convincing and provide some insights for the proposed method. I would like to increase my rating based on this.

---

### Official Review · Reviewer_X6YH · 2022-07-11

**Rating:** 7
**Confidence:** 5
**Soundness:** 4 excellent
**Presentation:** 4 excellent
**Contribution:** 3 good

**Summary:**

This submission shows how an approximate posterior distribution on DNN weights learned on a source task can be beneficial in transfer-learning to downstream tasks.

A 3-step method is proposed: 1) Fit posterior over weights on source task using SWAG. 2) Re-scale the learned posterior using a single scalar coefficient. 3) Plug the re-scaled prior into a Beysian inference algorithm like SGLD or SGHMC.

Experimental results are provided to support the following claims:

- In terms of performance, learned priors > SGD transfer learning > non-learned priors.
- SGD with learned priors significantly outperforms standard transfer learning (i.e., learned priors lead to performance gains with or w/o Bayesian inference).
- Learned priors lead to more data-efficient performance on the downstream task.
- When using the same prior Bayesian inference often outperforms SGD training -- though Bayesian inference with a non-learned prior has inferior accuracy to SGD with pre-training.


**Questions:**

**Q1.** What would be expected increase in computation (tuning included) w.r.t. conventional transfer learning for applying the proposed method to common tasks?

**Q2.** The method is compared to deep ensembles with regards to accuracy. How does the method compare with regards to overconfidence?


**Limitations:**

The two limitations discussed are the additional computation and the inclusion of only vision applications.

**Strengths And Weaknesses:**

# Strengths

**S1.** The method is well-motivated, presented with clarity and shown to lead to performance gains.

**S2.** The insights provided prompt further exploration of loss-surface alignment in transfer learning and other settings (e.g., few-shot learning).

**S3.** The method builds on prior art and adds a single hyperparameter. For this reason it should be easy for others to apply.

**S4.** The experimental validation includes classification tasks of different complexities and semantic segmentation. Thus, the method is shown to be beneficial in interesting problems.

# Weaknesses

**W1.** The proposed method claims to be a drop-in replacement for standard transfer learning.Because the additional computation is not expected to be trivial, it would have been beneficial to provide a proper study of the expected increase in computation w.r.t. standard transfer learning.

The submission includes some discussion about the additional training and inference cost but no discussion about the cost (in practice) of tuning hyperparameters. For example, there are a number of hyperparameters for the methods involved in the different steps of the proposed method. Setting or tuning these hyperparameters for a given transfer would be part of the cost of the method, in particular for steps 2 (scaling $\lambda$ of prior) and 3 (parameters of SGLD and SGHMC).

**W2.** The main technical novelty in the method is the scaling of the prior (steps 1 and 3 are applications of prior art) but this was not given much attention in the experiments (only Fig. 2c). What is the recommended procedure and expected cost for tuning the scaling parameter $\lambda$?

**W3.** Plots, legends and captions are not synchronized. For example, in Fig. 2c there is a purple plot but the legend does not include purple; in Fig. 7 the caption speaks of a red plot but there is no red in the plots.

---

> ### Author Response · Authors · 2022-08-02
> **Response to Reviewer X6YH**
>
> Thank you for your thoughtful and encouraging feedback!  We really appreciate that you recognized the simplicity of our approach as a strength, and found our presentation to be clear, and the method well-motivated and easy to apply, with a good evaluation. We note that in addition to this response post, we’ve also made a separate general post, with several clarifications and new results inspired by your comments. We respond to each of your points below:
>
> **W1 (computational cost):**  Inspired by your comments, we now provide a detailed discussion of computational costs in the general response to reviewers (see post titled “General Response to Reviewers and AC”) and in the paper, Appendix A, which we will move to the main body with the additional page allowed for the camera-ready version. In short, the computational overhead is typically minor and essentially the same as standard transfer learning if we do MAP optimization of the downstream posterior, which we show provides a significant performance boost over standard transfer learning.
>
> **W2 (procedure for the scaling parameter λ):**  In the general response, we now added several comparisons to alternative procedures, such as using the marginal likelihood, and layer-wise scaling parameters, out of consideration for your comments. We show that our original simple procedure provides essentially the same performance as these more sophisticated alternatives. We also note that our procedure is the same as tuning standard hyperparameters in transfer learning, requiring no specialized domain expertise, and performance is not sensitive to this parameter as in Figure 2d. Moreover, as we note in the general response, our tuning procedure adds only about 1/7th of the total runtime of standard transfer learning (including tuning other hyperparameters such as learning rate). For example, full hyperparameter tuning (including the re-scaling coefficient) and running MAP on the full Oxford Flowers-102 dataset takes 20 hours in total, whereas hyperparameter tuning and running traditional SGD-based transfer learning takes 18 hours on a computer with a single NVIDIA A100 GPU. We have added this discussion to Appendix A in the paper.
>
> **W3 (“Plots, legends and captions are not synchronized.”):**  Thank you for pointing out these formatting mistakes.  We have corrected each of these in our updated draft.
>
> In terms of vision applications, we wanted to go deep, rather than shallow and broad. We consider a variety of vision problems and performance across several dimensions, including the efficiency of learning on the downstream task, as well as a great performance on semantic segmentation and a careful ablation of each component of our procedure (especially with the updated results). Moreover, Bayesian inference itself in deep learning is mostly established for vision problems at this stage — we believe an exploration of entirely new domains would be most effective as a consideration in its own right, as a separate paper.
>
> **Q1 (“...expected increase in computation w.r.t. Conventional transfer learning…?”):**  We now address this in our response above and updated paper. We appreciate the question.
>
> **Q2 (“How does the method compare (to deep ensembles) with regards to overconfidence?”):**  We have now updated our draft to include test likelihood and reliability diagrams comparing our method with deep ensembles (see Figure 9 in the Appendix). In these experiments, deep ensembles achieve comparable results to BNNs without learned priors, in reliability diagrams and calibration, but worse results than BNNs with our learning priors.
>
> Thank you again for your thoughtful and supportive review. We made a significant effort to address your questions and would appreciate it if you would consider raising your score in light of our response.  Please let us know if you have additional questions we can address.
>
> ### References:
> [1] Maddox, Wesley J., et al. "*A simple baseline for bayesian uncertainty in deep learning*." Advances in Neural Information Processing Systems 32 (2019).
>
> [2] Immer, Alexander, et al. "*Scalable marginal likelihood estimation for model selection in deep learning*." International Conference on Machine Learning. PMLR, 2021.
>
> [3] Mackay, David John Cameron. "*Bayesian methods for adaptive models*." California Institute of Technology. 1992.

---

> > ### Author Response · Authors · 2022-08-06
> > **Response to Reviewer X6YH - Further engagement**
> >
> > Dear reviewer X6YH,
> >
> > Thank you again for your thoughtful review.
> >
> > Does our response address your questions?
> > We would appreciate the opportunity to engage further if needed.

---

> > > ### Comment · Reviewer_X6YH · 2022-08-08
> > > **Thanks for the additional details and experiments**
> > >
> > > Thank you for the additional details and experiments. I do find they strengthen the submission and will consider raising my rating.
> > >
> > > Please note I still find lack of synchronization between figures, captions and discussion. For example, re Fig. 9, the discussion mentions "a perfectly calibrated network has no difference between accuracy and confidence, represented by a dashed black line" but there is no such dashed black line. Also, the caption mentions error bars but there are no errors bars in the figure.

---

> > > > ### Author Response · Authors · 2022-08-08
> > > > **Response to Reviewer X6YH**
> > > >
> > > > Thank you for your response.
> > > >
> > > > We have now gone through each figure, caption, and corresponding description, and we have corrected inconsistencies in our updated draft.
> > > >
> > > > We are also happy that the new experiments strengthen the submission and that you will consider raising your score.

---

### Author Response · Authors · 2022-08-02
**General Response to Reviewers and AC - Part 1**

We thank the reviewers for their thoughtful and highly supportive reviews. We here provide a general response, addressed to all reviewers and ACs, as well as individual replies to address specific reviewer concerns as separate posts.

We want to emphasize the timeliness and significance of this work. Transfer learning is quickly becoming the de facto way of using deep learning models, but typically only transfers an initialization. We provide an effective pipeline that enables knowledge of the source task to influence the locations and shapes of the optima on the downstream task, a careful ablation of this pipeline, and a thorough evaluation, including self-supervised and supervised pre-training, showing detailed performance results as a function of datasize on several vision benchmarks, demonstrating sample efficient learning, and particularly strong results on semantic segmentation.  Modifying the loss surface on the downstream task via our informative priors leads to significant performance gains, with and without Bayesian inference. Bayesian inference provides a particular performance boost with the informative priors and also provides far superior calibration to standard transfer learning routines.

We also want to emphasize that the simplicity and modularity of this framework are key strengths. Our approach introduces only a single additional hyperparameter, which we show below is very easy to specify and does not require domain expertise or knowledge of Bayesian methods. Our informative priors (which we release to the public) are also compatible with MAP optimization (SGD), and our Bayesian inference routines require precisely the same amount of tuning, compute, and know-how as their standard Bayesian counterparts (without our priors).

Inspired by reviewer comments, we have now run several new experiments.  We present these experiments below along with a discussion of several points noted by reviewers:

### 1. Computational Complexity

We have added a detailed discussion of computational considerations to the updated version of our paper in Appendix A.

We evaluate the costs of each of the three stages of our pipeline: (i) inferring the posterior on the source task; (ii) re-scaling the posterior to become an informative prior for the downstream task; (iii) using the informative prior in the downstream task.

For (i), we note that we used SWAG because it has already been well-established as a tractable, well-studied, and simple way to infer a non-diagonal posterior, and it can be used starting from a pre-trained checkpoint [1]. We apply SWAG to a pre-trained checkpoint. We start with doing 50 iterations of burn-in and 10 cycles of 200 iterations each for a total of 2050 iterations with a batch size of 128. **In total, inferring the posterior costs only around a quarter of an epoch of training on ImageNet, and the computations in this step can be entirely avoided by users releasing pre-trained SWAG posteriors, as we do for all of the tasks in our paper.**

For (ii), we only need to tune a single re-scaling hyperparameter.  When we perform MAP optimization with our informative priors, tuning this scalar only adds 1/7 of the total runtime of transfer learning (including tuning other hyperparameters such as learning rate).  For example, the full hyperparameter tuning (including the re-scaling coefficient) and running MAP on the full Oxford Flowers-102 dataset takes 20 hours in total, whereas hyperparameter tuning and running traditional SGD-based transfer learning takes 17 hours in total on a machine with a single NVIDIA A100 GPU, which is a comparable amount of hyperparameter tuning to other works and significantly less than others [2,3]. We also note that the tuning is done with a simple validation grid-search, exactly the same way as any other hyperparameter tuning in standard transfer learning, and does not require any special expertise. Incidentally, this re-scaling hyperparameter is also a really simple and natural way to recognize that the source and downstream tasks come from different distributions — so, for example, a concentrated posterior on the downstream task does not provide an overly strong prior on the source task, and we show that it is practically effective.

---

> ### Author Response · Authors · 2022-08-02
> **General Response to Reviewers and AC - Part 2**
>
>
> For (iii), we offer two options, (a) MAP optimization of the downstream posterior using the informative prior, and (b) full Bayesian inference with SGLD or SGHMC using the downstream posterior. (a) **costs the same** as standard transfer learning both at train and test time, and we show it works better than standard transfer learning. (b) Bayesian inference with our informative priors bears the same cost as deep ensembles or as Bayesian inference with non-learning priors, which we compare to in our work.  A single chain of SGLD incurs the same cost as a single run of SGD, since the gradient of the log prior density can be computed without automatic differentiation in the same fashion as weight-decay, which we are replacing. We also compare multiple chains of SGLD with deep ensembles, which require the same cost, and show that Bayesian inference with our learned prior works significantly better.  Note that Bayesian inference and deep ensembles both have the same increase to the test-time cost, which we discuss in the main body of our original submission.
>
> In short, (i) has a minor cost (e.g., ¼ of an epoch on ImageNet) that is a one-time cost when a pre-trained posterior is publicly released, as we have done; (ii) involves a single hyperparameter that can be tuned in the same way as other hyperparameters, does not require specialized expertise, and adds 1/7 to the runtime of fine-tuning; (iii) has no additional cost if we do MAP optimization and has a cost comparable to deep ensembles if we run our implementation of Bayesian inference, both in terms of fine-tuning and test-time, and we show that Bayesian inference significantly outperforms deep ensembles.
>
>
>
> ### 2. Re-scaling the Prior
>
> As above, we introduce a single hyperparameter to rescale the posterior, as a simple, natural, and practically effective way to recognize that the source and downstream data come from different distributions, so that the informative prior is not over or under-concentrated on a particular setting of parameters.
>
> We tune this hyperparameter in exactly the same way as other hyperparameters in standard transfer learning, using a grid search on a held-out validation set, and as described above, it incurs minimal computational overhead. We stress that setting this parameter requires no specialized expertise. We also note that performance is stable over a large range of values, as we see in Figure 2(d), where performance is essentially the same over two orders of magnitude (10^5 to 10^7).
>
> More complex approaches for tuning priors exist. Inspired by reviewer comments, we have now implemented and run several of these. In particular, we now consider both per-layer [4] and single covariance scales, tuned using the Laplace marginal likelihood in [5], which requires approximate inference, but only training data, no validation data.
>
>
>
> Tuning a single covariance re-scaling hyperparameter using marginal likelihood didn’t improve the results reported in the paper on most of the datasets.
>
>
> Using different coefficients and re-scaling each layer achieved very slightly better results on some datasets. For example, the test errors on CIFAR-10, Oxford Flowers-102, and Oxford-IIIT Pets decrease by 0.03%-0.08%.
>
> Additionally, we also considered re-scaling the diagonal and low-rank components of the covariance matrix independently. This method, which requires tuning an additional hyper-parameter, achieves very slightly better results on most datasets. For example, the test errors on CIFAR-100, Oxford Flowers-102, and Oxford-IIIT Pets decrease by 0.03%-0.11%.
>
> We conclude that among numerous strategies for re-scaling the downstream posterior, our original approach of re-scaling a single coefficient with validation data is simple and effective, but some alternatives may achieve slight improvements in exchange for the additional computational and implementational costs. We have added a discussion of these points to Appendix D, and thank the reviewers for the suggestion.

---

> > ### Author Response · Authors · 2022-08-02
> > **General Response to Reviewers and AC - Part 3**
> >
> >
> > ### 3. Novelty and contribution
> >
> > As we opened, the paper makes several types of novel and significant contributions, and we  believe the simplicity of the pipeline is a strength, especially in light of the promising experimental results. In addition to the conceptual and practical contributions, as well as the timeliness of this paper, we would like to draw attention to:
> >
> > * Our extensive and successful evaluation on 6 datasets including classification and segmentation at a variety of sample sizes as well as multiple pre-training pipelines.
> >
> > * The carefully chosen yet simple design features which were important for making our pipeline work so well.  Our experiments show that the low-rank covariance component, SSL pre-training, and prior re-scaling are all essential for achieving high performance.  To further emphasize the importance of our specific design choices, we have now additionally run experiments using a Laplace approximation to the source posterior, a popular method for approximating posteriors [6] and found that SWAG performs significantly better. In fact, downstream Bayesian inference with the Laplace approximation prior performs worse than both SGD with learned (SWAG) priors and Bayesian inference with learned (SWAG) priors, and only slightly better than inference with non-learned priors in most cases (See full results in Tables 10-14 in the appendix).
> >
> > * Our exploration into why pre-trained priors work so well, after which we can confidently say that the pre-training loss surface geometry is both intuitively and in practice highly informative for downstream tasks as long as we account for the discrepancy by re-scaling the covariance matrix.
> >
> > ### References:
> > [1] Maddox, Wesley J., et al. "*A simple baseline for bayesian uncertainty in deep learning.*" Advances in Neural Information Processing Systems 32 (2019).
> >
> > [2] Li, Hao, et al. "*Rethinking the hyperparameters for fine-tuning*." International Conference on Learning Representations. ICLR, 2020.
> >
> > [3] Ting, Chen, et al. “*A Simple Framework for Contrastive Learning of Visual Representations.*” International Conference on Machine Learning. PMLR, 2020.
> >
> > [4] Mackay, David John Cameron. “*Bayesian methods for adaptive models.*” California Institute of Technology. 1992.
> >
> > [5] Immer, Alexander, et al. "*Scalable marginal likelihood estimation for model selection in deep learning*." International Conference on Machine Learning. PMLR, 2021.
> >
> > [6] Daxberger, Erik, et al. "*Laplace redux-effortless bayesian deep learning*." Advances in Neural Information Processing Systems 34 (2021)

---

### Meta-Review · Area_Chair_Djdg · 2022-08-31

**Recommendation:** Accept
**Confidence:** Certain

**Metareview:**

This work presents a Bayesian method for transfer learning using SWAG. Reviewers agree that this is well-motivated, it's novel and the proposed method is well done and works well. There are some concerns about the computational burden, but the authors claim that the proposed part adds about 1/7 total cost. I share some of the concerns with one of the reviewers regarding the fact that this method may be limited in usefulness to people who are experts, rather than the more general public. However, given that the method builds on prior art and adds a single hyperparameter, I feel it should be relatively easy for someone to actually use this, if they are interested in transfer learning.

**Award:**

No

---

### Decision · Program_Chairs · 2022-09-14

Accept